



# Minimizing the impact of vacating instream storage of a multi-reservoir system: a tradeoff study of water supply and empty flushing

**Chia-Wen Wu[1], Frederick N.-F. Chou[1], and Fong-Zuo Lee[2]**

[1] Department of Hydraulic and Ocean Engineering, National Cheng-Kung University, 1 University Rd., Tainan, Taiwan

[2] Hydrotech Research Institute, National Taiwan University, Taipei, Taiwan

Correspondence to: Frederick N.-F. Chou (chounf@gmail.com)

**Abstract**

Reservoir operator does not favor storage above a certain level in situations such as the pre-release operation prior to a flood, scheduled engineering constructions or mechanical excavations of sediments in the impoundments, drawdown and empty flushing, etc. This paper selects the last of which as the case study, and a method is presented to promote the

feasibility of emptying reservoir storage. The impact of emptying reservoir on water supply is minimized through appropriate joint operation in a multi-reservoir system, where drawdown and empty flushing is carried out in a primary reservoir and the other reservoirs provide backup water for supply. This method prioritizes allocating the storage in the primary reservoir for water supply during specific periods prior to its empty. If the storage of

every reservoir achieves its predefined conditions, drawdown of the primary reservoir is activated and followed by empty flushing. Previously preserved storage in the other reservoirs ensures adequate water supply during the periods of emptying the primary reservoir. Flushing of the primary reservoir is continued until either the accumulative released water exceeds the specified volume, storage in the backup reservoirs drops below

the pre-defined threshold, or the inflow to the primary reservoir recedes from the flood peak



to be below the releasing capacity of outlets. This behavior is simulated and linked with a nonlinear optimization algorithm to calibrate the optimal parameters defining the activation and termination of empty flushing. The optimized strategy limits the incremental water shortage within the acceptable threshold and maximizes the expected benefits of emptying

5 reservoir.

**Keywords***: water supply, joint operation of multiple reservoirs, drawdown of reservoir storage, empty flushing, reservoir desilting.





## 1. Introduction

Reservoirs intercept watercourses to store excessive water and regulate natural flow patterns into expected releases for different purposes. In contrast to conserve water, there are also circumstances in which storages above certain levels in reservoirs are not favored. These situations include when an imminent flood is anticipated, an extraordinary water quality is measured, engineering constructions or mechanical excavations of sediments within the impoundments are scheduled, and empty flushing operations. While the last of which is focused in this paper, the concept of minimizing the impact of vacating storages from reservoirs on the original purposes of water usage through careful management is general to all situations.

Empty flushing is the most effective method for removing deposited sediments from reservoirs (Fan and Morris, 1992; Morris and Fan, 1998; Shen, 1999). This process requires complete drawdown of reservoir storage to allow "inflows to pass through at riverine depths" (Atkinson, 1996). The drawdown of storage is usually carried out by releasing water through bottom outlets, such as sluiceways. During this process, the accelerated flow near the inlet may partially reactivate and scour out the depositions to generate a flushing cone in the vicinity of the inlet. By completely emptying the reservoir and maintaining the riverine flow condition, retrogressive erosion may be induced from the rim of the flushing cone extending to the upstream to create a flushing channel. The formation of the flushing channel usually leads to hyper sediment concentration of the bottom release and thus effectively recovers partial deposited capacity of the reservoir. This operation has been used to pursue sustainable utilization by many reservoirs worldwide (Atkinson, 1996; White, 2001; Chaudhry and Habib-ur-rehman, 2012), some examples of which are presented in Table 1.





Because draining the storage of a reservoir counteracts its water supply function, empty flushing is generally limited to reservoirs that operate solely for hydropower generation, flood mitigation, or irrigation. These purposes usually do not require reservoir storage during certain periods of the year, during which empty flushing can be implemented without impairing the original design function of the reservoir. However, for reservoirs with municipal or industrial water users that rely on sufficient storage for steady water supply, the implementation of empty flushing is relatively rare.

The conflict between water supply and empty flushing has been addressed by Chang et al. (2003) and Khan and Tingsanchali (2009). Chang et al. (2003) developed the operating rule curves and empty flushing schedule for the Dapu Reservoir in central Taiwan. A genetic algorithm was used to optimize the rule curves by minimizing the shortage index for irrigation while empty flushing is implemented only during prescribed periods. Khan and Tingsanchali (2009) applied a similar approach to the Tarbela Reservoir in Pakistan. The objective function in deriving rule curves is to maximize the net benefit from irrigational water supply, hydropower generation, sedimentation evacuation and flood mitigation. However, these previous studies dealt only with single-reservoir. If there are additional reservoirs in the system that can act as backup water sources, it may be possible to elevate the feasibility of empty flushing by reducing its impact on water supply through appropriate joint operation. The ideal strategy may require utilizing the reservoir with the most excessive sediment deposition, referred to as the primary reservoir throughout the remainder of this paper, to supply demands while preserving the storage in the other reservoirs before empty flushing. This will lead to a lower water surface level (WSL) in the primary reservoir and a higher WSL in the others. Empty flushing in the primary reservoir can then be activated once favorable conditions are achieved, such as adequate storage distribution among reservoirs to ensure both high sediment flushing efficiency and steady backup water supply.



This study focused on a water resources system that contains multiple reservoirs, among which one primary reservoir requires appropriate empty flushing operation while the others could provide backup storage. The goal is to develop the optimal strategy, which maximizes the efficiency of empty flushing, for such a system. In the following section, key factors influencing the efficiency of sediment flushing as well as the stability of water supply are discussed. The methodology to derive the strategy for joint water supply operation and empty flushing in a multi-reservoir system is then presented. The proposed approach is applied to a tandem online and offline reservoir system in southern Taiwan. The results from the case study validate the efficacy of the derived optimal strategy.


## Table 1 Cases of empty flushing

| Reservoir | Country | Effective capacity (Mm3) | Major Purpose | Number of flushing days per operation | Reservoir system | Watershed sediment yield (M.ton/year) | Capacity-inflow ratio (CIR) | Annual flushing periods | Backup water supply during flushing | Flushing facilities | Flushing experiences | Literature source |
|---|---|---|---|---|---|---|---|---|---|---|---|---|
| Akung-ten | Taiwan | 16.47 | FC IR | 100 days | S | 0.55 | 0.71 | June to September | Trans-basin diversion from an adjacent river | morning glory with capacity of 85 m³/s | Flushing out 5% to 54% of the inflowing sediments during floods between 2009 to 2013 | Southern Water Resources Office, 2013 |
| Baira | India | 2.40 | HP | 1 to 2 days | S | 0.3 (from siltation after 18 months) | 0.001 | April to May | Halting hydro-power generation | low–level diversion tunnel with capacity of 44 m3/s | The first operation in Aug of 1983, lasting for 40 hours with discharge of 44 m3/s, flushed out 85% of the deposited sediments. Afterwards the empty flushing is suggested to be annually performed during April to May. | Jaggi and Kashyap, 1984; Atkinson, 1996; Chaudhry and Habib-ur-rehman, 2012 |
| Cachi | Costa Rica | 54.0 | HP | 2 to 3 days | S | 0.81 | 0.016 | May (the beginning of wet season) | Halting hydro-power generation | Bottom outlet | On average flushing out 0.25 million m3/year of sediments annually. | Brandt and Swenning, 1999; Jansson and Erlingsson, 2000; Chaudhry and Habib-ur-rehman, 2012; |
| Dapu | Taiwan | 5.29 | IR ID | 50 days | S | 0.40 | 0.04 | May to July | Halting irrigational supply. Industrial demand is supplied by reservoir inflow | Sluiceway with capacity of 325 m3/s | On average flushing out 0.20 million m3/year of sediments annually. | Chang et al, 2003; Water Resources Agency, 2010 |
| Gebidem | Switzerland | 9.0 | HP | 2 to 4 days | S | 0.50 | 0.02 | May to June | Halting hydro-power generation | Bottom outlet with flushing discharge of 10 m3/s | Since 1992, the annual volume of flushed sediments is between 0.2 to 0.5 million m3 per year. | Atkinson, 1996; Chaudhry and Habib-ur-rehman, 2012; Meile et al., 2014 |

HP: hydropower generation, FC: flood control, IR: irrigation, ID: industrial water supply, S: single reservoir system, M: multi-reservoir system,

Capacity-inflow ratio: the ratio between the effective capacity and the annual inflow volume of the reservoir,



## Table 1 Cases of empty flushing (continued)

| Reservoir | Country | Effective capacity (Mm3) | Major Purpose | Number of flushing days per operation | Reservoir system | Watershed sediment yield (M.ton/year) | Capacity-inflow ratio (CIR) | Annual flushing periods | Backup water supply during flushing | Flushing facilities | Flushing experiences | Literature source |
|---|---|---|---|---|---|---|---|---|---|---|---|---|
| Heng-shan | China | 13.30 | FC IR | 10 to 20 days | S | 1.18 | 0.84 | June to September | Diverting turbid release to provide irrigational demand. | Bottom outlet with capacity of 17 m³/s at full impounding level and 2 m³/s during empty flushing | The first operation in 1974 lasted for 37 days and flushed out 0.8 million m³ of sediments. The second operation in 1979 lasted for 52 days and flushed out 1.03 million m³ of sediments. | Atkinson, 1996; Chaudhry and Habib-ur-rehman, 2012 |
| Jensan-pei | Taiwan | 1.51 | IR | 53 days | S | 0.28 | 0.80 | May to June | Halting water supply | Sluiceway with capacity of 12.2 m³/s | On average flushing out 0.33 million m³/year of sediments annually. | Water Resources Planning Institute, 2010 |
| Manga-hao | New Zealand | 2.39 | HP | 30 days | M | – | – | May | Halting hydro-power generation | low–level diversion tunnel | During the total duration of one month of flushing in 1969, 0.8 million m³ of sediment has flushed from the reservoir, which equals to the 75% of sediment that had accumulated since 1924 | Jowett, 1984; Atkinson, 1996; White, 2001; Chaudhry and Habib-ur-rehman, 2012 |
| Nan-qin | China | 10.20 | IR FC | 4 days every 3–4 years | S | 0.53 | 0.08 | The end of the flood season | --- | Sluiceway with flushing discharge of 14 m³/s | The first operation in 1984 flushed out all inflow sediments in 1984, along with 0.72 million m3 that had deposited in the earlier years. | Chen and Zhao, 1992; Chaudhry and Habib-ur-rehman, 2012 |
| Santo Domingo | Vene-zuela | 3.00 | HP | 3 to 4 days | S | 0.20 | 0.01 | May | Halting hydro-power generation | Three bottom outlets with capacity of 13 m³/s | The first operation in May of 1978 lasted for 4 days and flushed out 50–60% of the deposited sediments. Afterwards mechanical excavation was used to disperse the consolidated deposits and empty flushing is again performed for three weeks to fully restore the deposited capacity. | Krumdieck and Chamot, 1979; Atkinson, 1996 |

HP: hydropower generation, FC: flood control, IR: irrigation, ID: industrial water supply, S: single reservoir system, M: multi-reservoir system,

Capacity-inflow ratio: the ratio between the effective capacity and the annual inflow volume of the reservoir,





**Table 1 Cases of empty flushing (continued)**

| Reservoir | Country | Effective capacity (Mm3) | Major Purpose | Number of flushing days per operation | Reservoir system | Watershed sediment yield (M.ton/year) | Capacity-inflow ratio (CIR) | Annual flushing periods | Backup water supply during flushing | Flushing facilities | Flushing experiences | Literature source |
|---|---|---|---|---|---|---|---|---|---|---|---|---|
| Sefid-Rud | Iran | 1760 | IR HP | 4 months | S | 50 | 0.36 | October to February | No requirement for irrigational water supply | Bottom outlets with flushing discharge of 100 m³/s | Empty flushing during non-irrigational periods removes approximately 28.4 million T of sediments per year. | Atkinson, 1996; Taklimy and Tolouie, 2005 |
| Zemo-Afchar | Former USSR | -- | HP | 1 to 3 days | S | Suspended load 4 Mm³ | -- | April, May or November | Halting hydro-power generation | Bottom outlets with flushing discharge of 450 m³/s | Implemented from 1939, with full drawdown. Removing about 1.0 million m³ (from 0.5 to 2 million m³) per year | Bruk, 1985; Chaudhry and Habib-ur-rehman, 2012; |
| Dashidaira | Japan | 1.657 | HP FC | 1 to 2 days | M | 0.62 | 0.00674 | June to August | Halting hydro-power generation | Bottom outlets with flushing discharge between 200~300 m³/s | When inflow at the upstream Dashidaira Dam exceeds 300 m³/s at the first time of the year during June to August, a coordinate flushing is performed. The average annual flushed volume between 2001 to 2007 is 0.27 million m³/year | Sumi, 2008; Sumi et al., 2009 |
| Unazuki | | 12.70 | | | | 0.96 | 0.014 | | | | | |
| Verbois | Switzerland | 12.00 | HP | 1 to 2 days every 3 years | M | 0.33 | 0.00144 | May to June | Halting hydro-power generation | Bottom outlets with flushing discharge of 600 m³/s | Flushing is performed in every 3 years. The volumetric flushed sediments per event is around 0.6 and 1.1 million m³ for Verbois Reservoir and 0.1 and 0.4 million m3 for Genissiat Reservoir according to Sumi (2008) | Sumi, 2008 |
| Genissiat | France | 18.00 | | | | 0.73 | 0.00467 | | | | | |

HP: hydropower generation, FC: flood control, IR: irrigation, ID: industrial water supply, S: single reservoir system, M: multi-reservoir system,

Capacity-inflow ratio: the ratio between the effective capacity and the annual inflow volume of the reservoir,


## 2. Material and Methods

### 2.1 Qualitative analysis: Key factors for successful operations of empty flushing

Two major performance indices, expected desilting volume and the induced increments of water shortage, are used in this study to evaluate an empty flushing strategy. An optimal strategy should maximize the desilting volume while maintaining the incremental shortage under an acceptable threshold. According to the cases in Table 1, key hydrological and operational factors for succeeding in these indices are identified as follows:

1. Qualitative conditions for water supply (QCWS)

    (1) QCWS 1: Adequate water supply *during* empty flushing

    In order to satisfy this condition, episodes between periods with heavy water supply pressure can be utilized as windows of opportunity to implement empty flushing. One example is the Dapu Reservoir in central Taiwan, which primarily provides agricultural and industrial water supply. Empty flushing of this reservoir is scheduled from May to July when irrigation demand is low and reservoir inflow is sufficient for demands. In contrast, reservoirs that provide water to the general public must maintain at stable supply level throughout the year. Empty flushing of such reservoirs would require backup water sources capable of ensuring a steady supply until the reservoirs can resume normal operations. One example is the Agongdian Reservoir in southern Taiwan which undergoes empty flushing from June to September annually, during which trans-basin diversion from an adjacent basin adequately supplements public and agricultural water supply.

    (2) QCWS 2: Adequate water supply *after* empty flushing

    Satisfaction of this condition requires sufficient reservoir inflow following empty flushing to rapidly replenish the storage of the reservoir. Thus, the capacity of a



reservoir undergoing empty flushing is usually relatively small compared to the volume of its inflow. Basson and Rooseboom (1997) indicated that empty flushing is more feasible for reservoirs with an effective capacity to annual inflow volume ratio (capacity-inflow ratio, CIR) of less than 0.03. Many of the reservoirs in Table 1 fulfill this criterion. The others that have a CIR greater than 0.03 are located in areas with uneven seasonal rainfall distributions, such that the abundance of inflow during flood seasons can effectively refill the storage soon after empty flushing. One example is the Dapu Reservoir, which receives abundant rainfall and inflow from May to August every year. This particular reservoir can remain empty until early July without affecting the subsequent irrigational water supply.

2. Qualitative conditions for flushing sediments (QCFS)

Compliance with these conditions promotes efficiency of sediment flushing. The key is to identify and take advantage of opportunities with both high inflow and low WSL of the reservoir to perform empty flushing.

(1) QCFS1: High inflow during empty flushing

High inflow is required to maximize the flushing efficiency by more effectively scouring the depositions of the reservoir. Atkinson (1996) and White (2001) indicated that empty flushing should only be initiated when the inflow is at least double the inflow in normal conditions. The experience with flushing the Zemo-Afchar Reservoir of the former USSR (Chaudhry and Habib-ur-rehman, 2012) suggests that empty flushing is most effective with inflow between 400 to 500 $m^3$/s, which is 2 to 2.5 times the average inflow (Bruk, 1985; Singh and McConkey-Broeren, 1990). The operating experiences of Jianshanpi Reservoir in southern Taiwan also suggest that the efficiency of empty flushing peaks during heavy rainfall events when daily rainfall on the reservoir watershed is between 40 to 60 mm. This condition can also be artificially



achieved. For instance, during the empty flushing of the Mangahao Reservoir in New Zealand, water was released from another upstream reservoir to enhance the scouring of depositions and thus maximize desilting volume (White, 2001).

(2) QCFS2: Low WSL *before* and *during* empty flushing

    a. *Before* empty flushing is started: operators could take advantage of periods when the reservoir WSL is low to perform drawdown and initiate empty flushing. In cases where the reservoir has outlets with sufficient capacities, timely drawdowns can be performed shortly prior to expected floods so that the flood inflow can effectively scour and flush out depositions. One example is the Dapu Reservoir, of which WSL is generally the lowest in mid-May. This timing is thus considered as the ideal time to empty the reservoir, with the expectation that subsequent abundant floodwater from May to August can also fulfill the QCWS 1, QCWS 2 and QCFS 1.

    b. *After* empty flushing is initiated: Once empty flushing is initiated, the reservoir should remain as close to empty as possible to maintain high flushing efficiency. However, if the inflow exceeds the capacities of the outlet works, then the WSL in the reservoir will begin to rise. This leads to decreased flow velocity in the reservoir, which reduces the empty flushing efficiency. Atkinson (1996) suggested the use of the drawdown ratio (DDR) to measure the flushing efficiency. This index is defined as 1 minus the ratio between the depth of WSL during empty flushing and the depth of normal pool level of the reservoir. Atkinson (1996) and White (2001) defined incomplete drawdown flushing as situations in which DDR is less than 0.66, wherein the depth of the water during flushing is greater than a third of the maximum depth. In such circumstances, the efficiency of empty flushing is significantly reduced and it is recommended to switch the operation to the regular mode of water supply.





## 2.2 Quantitative derivation of the optimal empty flushing strategy

While the discussion of the previous subsection is generally applicable for any multi-reservoir systems, the proposed quantitative methodology as well as the following case study apply specifically to those without means to artificially generate flushing inflow to the primary reservoir. In addition, we focus on event-based operation. This means that the timing and duration of empty flushing is flexible according to real-time hydrological and operational conditions. If these conditions are not favorable, the primary reservoir could resume regular operation of water supply. This feature distinguishes the present method from previous related studies (Chang et al., 2003; Khan and Tingsanchali, 2009), which mandatorily empty reservoir storage during predetermined periods within a year. This paper also assumed that the water demands in the system require constant supply, thus rendering empty flushing infeasible during parts of the year. To facilitate determination of feasible periods for empty flushing, the following criteria are provided:

1.  Meeting QCWS 1 requires periods of low water demand during which backup reservoirs can provide adequate supply during empty flushing.

2.  QCFS 2 dictates that the most opportune time to begin empty flushing is at the end of the dry season. At this time the storages of reservoirs are usually at their lowest levels of the year. This ensures that storage can be effectively and efficiently drained by drawdown flushing through the capacitated bottom outlets of the primary reservoir.

3.  Meeting QCFS 1 requires that the feasible duration for empty flushing should be extended into the wet season to ensure adequate inflow for scouring depositions.

4.  Meeting QCWS 2 requires that the feasible duration for empty flushing should be ended before the end of the wet season to ensure adequate replenishment of reservoir storage after the flushing operation.



The following proposed method for deriving optimal strategy adopts the simulation-optimization linkage approach. It requires simulating the operations of water supply and empty flushing, thus allowing for quantifying the desilting volume as well as the incremental water shortage generated by a given strategy. The process of water supply is simulated according to a set of joint operating rules as presented in subsection 2.2.1. When specific quantitative conditions presented in subsection 2.2.2 are achieved, empty flushing in the primary reservoir is activated and the approach in subsection 2.2.3 is employed to estimate the desilting volume. The empty flushing terminates when the conditions presented in subsection 2.2.4 are reached, and the simulation is switched to regular water supply operation until the next time activation conditions are satisfied. The simulation model is linked to an optimization algorithm to calibrate optimal parameters in the activation and termination conditions, according to the formulation presented in subsection 2.2.5. Fig. 1 depicts a flowchart of the analyzing procedure.





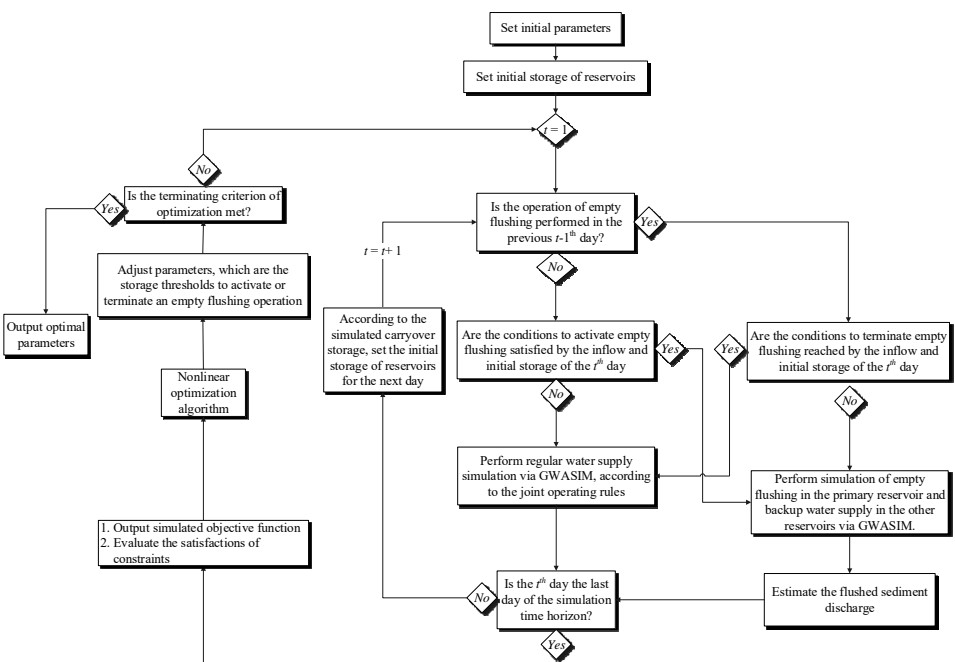

**Fig. 1 The procedure to derive the optimal empty flushing strategy**

### 2.2.1 Joint operating rules for a multi-reservoir system

According to Oliveria and Loucks (1997), the rules to jointly operate multiple reservoirs for water supply include the following two phases:

1. Determination of total water supply amount: The total amount of water supply is determined based on the total storage of reservoirs. If the total storage does not suffice, a discount of total water supply may be applied by the system-wide release rule. Fig. 2

presents the joint operating rule curves, a form of the system-wide release rule, for the Tsengwen and Wushanto Reservoirs in southern Taiwan. The location and associated water resources system of these reservoirs are depicted in Figs. 4 and 6 in the case study section. The release rules stipulate that when the total storage of the two reservoirs is below the critical limit, only 80% of the public demand and 50% of the agricultural and

industrial demands will be satisfied. When the total storage is between the lower and



critical limits, the public demand should be fulfilled and 75% of the agricultural and industrial demands need to be satisfied. When the total storage is between the upper and lower limits, all demands should be fulfilled. In the event that the storage in the Tsengwen Reservoir exceeds the upper limit, extra water can provide excess supply or full loaded

hydropower generation until the storage returns to the upper limit.

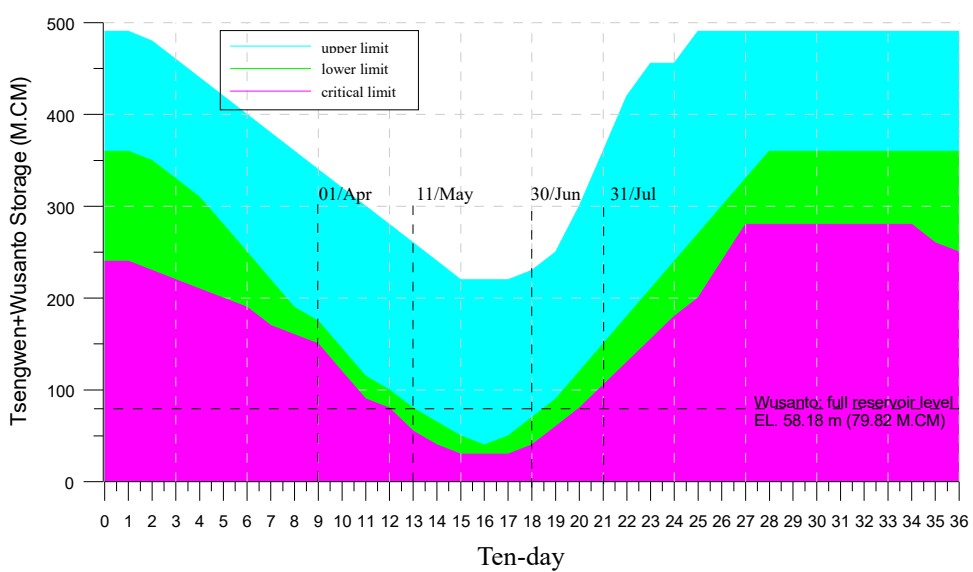

**Fig. 2 Joint operating rule curves of the Tsengwen and Wushanto Reservoirs**

2.  Distributing storage to individual reservoirs: Based on the calculated total water supply, the total end-of-period storage in the system can be estimated with the expected reservoir inflow during one single operating period. The release from each individual reservoir can then be determined by applying an individual reservoir storage balancing function, such as storage balancing curves. Fig. 3 exhibits the storage balancing curves for the Tsengwen

and Wushanto Reservoirs in early April (Southern Regional Water Resources Office, SRWRO, 2012). The horizontal axis in the figure measures the total storage in the system,



and the two curves represent the suggested target storages for the respective reservoirs with regard to various total storage amounts. These curves vary during each ten-day period within a year to facilitate efficient storage allocation according to the pattern of water demands and reservoir inflow.

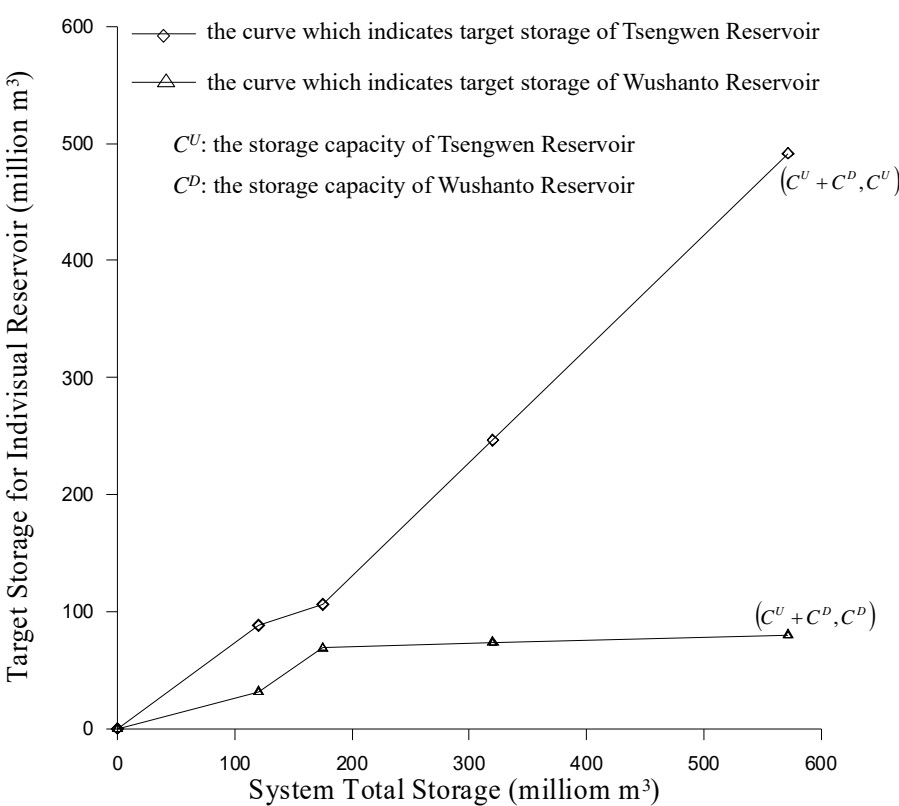

**Fig. 3 Storage balancing curves for Tsengwen and Wushanto Reservoirs in the tenth ten-day period (early April)**

Fig. 3 was designed to ensure efficient utilization of water resources and adequate

10     supply during which the agricultural demand peaks, without any consideration of empty flushing. The first part of the proposed method requires appropriate adjustment of the storage balancing curves before and during the periods feasible for empty flushing. This adjustment





would prioritize the water released from the primary reservoir while preserving storage in the other. This complies with the aforementioned QCWS1 and QCFS2, and creates a favorable initial condition for empty flushing.

**2.2.2 Conditions for initiation of an empty flushing operation**

Water supply simulation of historical daily reservoir inflow records is sequentially performed according to the joint operating rules. During the simulation, empty flushing operation is activated when all of the following conditions are satisfied:

1.  The current simulating date falls within the pre-evaluated feasible timeframe for empty flushing.

2.  The storage of the primary reservoir is lower than a threshold $T^U$. This ensures the satisfaction of QCFS2. A higher value of $T^U$ allows initiating drawdown flushing at higher primary reservoir storage levels, thus increasing the range of opportunities for empty flushing. Nonetheless, a higher $T^U$ incurs the risk that, if subsequent reservoir inflow falls short of predicted values, the emptied storage may not be replenished.

3.  The total storage in the backup reservoirs is greater than a threshold $T^D$. This ensures meeting QCWS1. A higher value of $T^D$ elevates the stability of water supply during empty flushing. In cases where either this or the above condition has not been met, demand should be supplied from the primary reservoir as much as possible, or storage should be diverted from the primary reservoir to the others. However, this storage

reallocation may be limited by the water transmitting capacities between reservoirs. Such that the conditions for initiating empty flushing may not be met within the pre-specified



feasible period for flushing. Therefore, a higher $T^D$ may reduce the opportunities to perform empty flushing.

### 2.2.3 Estimation of the flushed sediment discharge

Once the activation conditions are met, the gates of the bottom outlets of the primary reservoir are fully opened to empty the storage and route the inflowing water and sediments. The release from the primary reservoir may cause blockages of the downstream water diversion or water treatment facilities due to its high sediment concentration. Thus the water supply may rely solely on the storage preserved in the other reservoirs. During empty flushing, the inflow, outflow and WSL of the primary reservoir can be used to estimate the volume of flushed sediments. The estimation can be based on either numerical simulation or empirical formula, the second of which is adopted by this paper due to its simplicity and easy incorporation with the proposed optimization framework. After the optimized strategy identifies feasible events of empty flushing, numerical simulation is then used to verify the effectiveness of the estimation by empirical formula.

The empirical formula developed by the International Research and Training Center on Erosion and Sediment (IRTCES) in Tsinghua University, Beijing (IRTCES, 1985) is employed for the estimation of releasing sediment discharge from the primary reservoir. The formula is based on measurements from 14 reservoirs in China:

$$QC_t = \psi \frac{Q_t^{1.6} S_f^{1.2}}{W^{0.6}}$$

(1)

where $QC_t$ and $Q_t$ denote the sediment discharge (T/s) and water discharge (m$^3$/s) flushed from the primary reservoir during the t-th simulating day, respectively; $S_f$ represents the energy slope associated with the flow in the primary reservoir during empty flushing; $W$ is the width of the flushing channel (m), which can be estimated using the empirical formula



$W = 12.8 \cdot Q^{0.5}$ (Atkinson, 1996), and $\psi$ is the flushing coefficient, associated with the characteristics of the sediment and topography of the reservoir.

### 2.2.4 Conditions for termination of empty flushing operation

Empty flushing operations should be terminated if one of the following circumstances occurs:

1. The accumulative released volume from the primary reservoir exceeding a certain threshold $T^V$ would halt the flushing of the year. Empty flushing consumes water originally stored in the reservoir for water supply. This consumption is expected to be compensated by subsequent floods which could fulfill the reservoir and induce spillage. Nonetheless, if the floods are not significant enough following the flushing, the impacts of excess releasing will be carried on to the next dry season and increase water shortage. The threshold $T^V$ alleviates this impact by restraining the water consumption volume of empty flushing.

2. Providing the accumulative release volume is still under $T^V$, the flushing should be terminated when the flood flow has raised the WSL of the primary reservoir and daily inflow subsequently recedes to be below the capacity of associated bottom outlets. This situation indicates that the operation has been successfully timed to encounter a flood and should thus be ended when the flood ends.

3. The flushing should be ended when the storage of backup reservoirs decreases to below a threshold $T^d$. This condition prevents short-term water shortages following flushing operations resulting from insufficient storage. During the flushing operation, the primary reservoir will remain empty in the absence of floods, so providing water supply will gradually reduce available storage in the other reservoirs. A higher value of $T^d$ will cause the storage below threshold more quickly, thus reducing the window of operation for



empty flushing. Nonetheless, adequate reservoir inflow and proper storage reallocation after an earlier termination of one flushing operation will facilitate the re-initiation of a subsequent operation during the feasible period for empty flushing. Thus, under conditions of a higher $T^d$ value, the pattern of empty flushing may be transformed from a few operations of longer duration into multiple intermittent operations of shorter durations.

A generalized water allocation simulation model (GWASIM) developed by Chou and Wu (2010) is used to simulate the alternating operations of empty flushing and joint water supply according to the aforementioned rules and conditions. The structure of this model is formulated in network flow programming. It has already been implemented in the planning and management of all major water resources systems in Taiwan. Details of its simulations regarding the operations of multi-reservoir systems can be found in Chou et al. (2006) and Chou and Wu (2014).

### 2.2.5 Evaluation of optimal empty flushing strategies

The thresholds for activating and terminating an empty flushing operation as described in subsections 2.2.2 and 2.2.4 are regarded as parameters. These parameters are calibrated to maximize the total desilting volume without inducing both short- and long-term intolerable water shortage scenarios. The short-term scenario assumes the occurrence of subsequent floods cause reservoir spillage and fully compensate for the impact of emptying reservoir. Thus the incremental shortage following empty flushing is concentrated in a few months before the consequent floods, during each of which the monthly shortage increment and ratio is calculated respectively:

$$d_{n,m}^{I} = d_{n,m} - d_{n,m}^{0} , \quad m = 0,1,2,...,n^m, \ n = 1,2,...,n^y \qquad (2)$$





$$d_{n,m}^{R} = \frac{d_{n,m}}{D_m}, \quad m = 0,1,...,n^m, \ n = 1,2,...,n^y \tag{3}$$

where $d_{n,m}^{I}$ and $d_{n,m}^{R}$ represent the water shortage increment and ratio during the $m$-th month following the feasible period of empty flushing in the $n$-th simulating year; $D_m$ denotes the water demand during the $m$-th month following empty flushing; $d_{n,m}$ and $d_{n,m}^{0}$ represent simulated water shortages under conditions with and without empty flushing operations. $d_{n,m}^{0}$ is from simulating the default regular water supply process using the GWASIM and $d_{n,m}$ is obtained by incorporating empty flushing operations according to the activating and terminating conditions defined by the parameters. $n^m$ is the number of months within which the impact of empty flushing on water supply is carried over; and $n^y$ is the number of simulating years.

The long-term shortage scenario aims at situations that subsequent floods are not significant enough to induce reservoir spillage. Thus the incremented shortage extends into the next dry season, of which shortage ratios with or without empty flushing are expressed as below:

$$d_{n,dry}^{R,0} = \frac{d_{n,dry}^{0}}{D_{dry}}, \quad n = 1, 2, ..., n^y \tag{4}$$

$$d_{n,dry}^{R} = \frac{d_{n,dry}}{D_{dry}}, \quad n = 1, 2, ..., n^y \tag{5}$$

where $D_{dry}$ represents the total demand during the dry season, $d_{n,dry}^{0}, d_{n,dry}, d_{n,dry}^{R,0}$ and $d_{n,dry}^{R}$ are the water shortage volumes and ratios with and without executing empty operation during the dry season, respectively.



The formulation of the optimization problem is as follows:

$$Maximize \quad \sum_{t=1}^{n^t} QC_t \tag{6}$$

subject to

$$\max_{n=1,...,n^y}(d_{n,m}^R \mid d_{n,m}^I > 0) \le \alpha \quad m = 0,1,2,...,n^m \tag{7}$$

$$\max_{n=1,...,n^y}(d_{n,dry}^R \mid d_{n,dry}^{R,0} \le \beta) \le \beta \tag{8}$$

where $n^t$ is the total number of days within the simulating horizon; $QC_t$ is the simulated sediment discharge from the primary reservoir by empty flushing on the t-th day. It is determined by substituting the release of the primary reservoir during the flushing period into Eq. (1). The left hand side (LHS) of Eq. (7) represents the maximum water shortage ratio of

10 the m-th month from the $n^y$ simulating years given that the short-term shortage is induced by empty flushing. The right hand side (RHS) of Eq. (7), $\alpha$, is the maximum acceptable shot-term monthly shortage ratio induced by empty flushing. The LHS of Eq. (8) represents the maximum shortage ratio during the dry season following empty flushing, given that the original shortage ratio of the same periods without flushing is less than a failure threshold.

The failure threshold of shortage ratio $\beta$ usually means the irrigation for majority areas needs to be suspended to ensure steady public water supply in Taiwan. Eq. (8) thus avoids inducing additional failure events of water supply in the dry season following empty flushing.

The BOBYQA, a nonlinear optimization algorithm of Powell (2009), is used to solve the problem. The details of BOBYQA can be found in Powell (2009) and the barrier function

approach to handle the constraint of Eqs. (7) and (8) can be found in Chou and Wu (2015).

**2.3 Alleviating the impacts on downstream environment**





In addition to induce incremental water shortage, potential adverse impacts on the downstream environment may appear due to flushing. They originate from the high sediment concentration of flushing release in the absence of a simultaneous downstream flood, which may significantly reduce dissolved oxygen in the river, smother stream benthos, clog gravels

thus endangering spawn sites and habitat, impairing river functions such as flood conveyance, navigation or recreation, etc (Morris, 2014). To minimize these impacts, Morris (2014) suggests to carefully determining the timing, duration and frequency of empty flushing. The general principle is to limit the volume and duration of the flushed sediments detained on the downstream river bed. This leads to the following measures:

1. To identify and avoid periods during which stream creatures are sensitive to water quality.

    2. To schedule the empty flushing while natural floods downstream or releases from other reservoirs are available during the same storm to dilute and transport sediments.

    3. To ensure that subsequent high-streamflow, either from tributaries or reservoir releases, is available to clear the accumulated sediments from previous flushing events.

4. Adopting more frequent, periodical and short period of flushing to attenuate the release concentration in each operation.

This paper focuses on balancing water supply and empty flushing of sediment through proper joint operation of reservoirs. The role of securing water supply acts similarly as mitigating environmental impacts, since they both demand to restrain the effectiveness of

empty flushing. The impacts to these two purposes are addressed as below:

    1. Meeting QCWS1 and QCFS1 suggests the flushing periods to coincide with the first flood of wet season, which is usually induced by the frontal plum rain in Taiwan, while subsequent typhoon-induced floods are expected to satisfy QCWS2. These conditions conform to the principles of limiting environmental impacts, since they prompt adequate

water during and following flushing for dilution and transportation of sediments



downstream.

2. While the first termination condition in subsection 2.2.4 limits the releasing volume for reservoir bottom outlets, the second condition also allows empty flushing until the inflow recedes below the capacity of bottom sluiceway of the primary reservoir. The release during the recession limb of flood inflow with lower concentration and higher WSL of reservoir also partially offsets the impact immediately following the empty flushing.

3. As for the third termination condition, the strict requirement on the stability of water supply leads to a higher $T^d$, the threshold storage in backup reservoirs to stop an empty flushing operation. In the absence of a flood, a higher $T^d$ induces an earlier termination and increases the opportunities of re-activation of another empty flushing operation. This results in more frequent flushing with shorter duration, which is considered as more favorable to mitigate downstream impacts (Morris, 2014; Sumi and Kantoush, 2010; Crosa *et al*., 2010).

4. Some characteristics of the case study area regarding the impact from empty flushing on downstream environment are addressed as below:

    (1) The bottom outlet available for empty flushing of the primary reservoir, Twengwen Reservoir, is its permanent river outlet (PRO) with design capacity as 177 m³/s. Its actual capacity when the storage is nearly empty decreases to around 130 m³/s. According to the operating guidelines of Twengwen Reservoir, the minimum operating spillway release is 300 m³/s and a release under 2,250 m³/s is considered as free of inducing downstream flooding damage. It shows that release from empty flushing is relatively insignificant comparing to the magnitudes of spillway releases and downstream lateral flow during a moderate flood.

    (2) The current operators adopt dredging, or more precisely hydrosuction, as the major approach to desilt the Twengwen Reservoir. Several soil dikes were constructed in



the river immediately downstream from the dam as temporary depositing area for the dredged slurries. The lump-sum volume of the depositing area which defines the annual dredging capacity is currently 2.8 million m$^3$ and to be expanded to 3.5 million m$^3$ in the future. The artificially deposited sediments as well as the soil dikes

are expected to be flushed out by the spillway release during typhoons to allow continuation of hydrosuction in the next year. The required volume from the reservoir release to vacate downstream depositions and soil dikes is estimated as 40 million m$^3$ (900 m$^3$/s times 12 hours) based on field operating experiences. Previous surveys (Water Resources Agency, 2018) showed that these operations have limited

impacts on the flood conveyance function as well as the weighted usable area of a major interested benthos specie in the downstream river. It is expected the sediments from empty flushing during the first flood can be partially detained in the depositing area, and then carried downstream along with the dredging depositions by the subsequent flood operation during typhoons in the rest of wet season.

(3) The above mentioned benthos specie of interest is Sinogastromyzon nantaiensis (Chen et al., 2002). Field investigation showed its spawning season is from June to September, and its habitation site is within the first downstream reach from the Twengwen Reservoir. The most threat for the benthos is the fine suspended or even wash loads from either hydrosuction, flushing or sluicing operations, which may fill

up the voids of river bed and destroy its spawning sites. Hence the empty flushing operation is limited before mid-June to minimize the concurrence periods of the spawning season. Also the first river reach downstream from the dam has a moderate slope as 0.0036 and short distance as 20.5 km. These prompt transportation of sediments by the spillway release during typhoons.

(4) If, however, typhoons are absent or not strong enough during the wet season





following empty flushing, spillway release may be inadequate to fully vacate downstream depositions. A possible counteraction may require deliberately releasing water through spillway at the end of wet season to not only compensate the impact of empty flushing but recover depositing volume for hydrosuction. This will inevitably increase the water shortage risk in the following dry season and require water saving measures from the irrigation stakeholders.

In the following section of result and discussion, volumes of downstream lateral inflow during the empty flushing and spillway release during the rest of wet season are also listed for each identified event to address the impact of empty flushing on environment. Suggestions for incorporating numerical simulations of downstream sediment transport with the operation of reservoir are also provided for future studies.

## 2.4 Case study and experimental setup

The joint operating system of the Tsengwen and Wushanto Reservoirs in southern Taiwan is selected for case study. Fig. 4 shows the location of these reservoirs. The Tsengwen Reservoir is in the upper section of the Tsengwen River, with a watershed area of 481.6 km$^2$. The original effective capacity with the WSL as the normal pool level El. 227 m was 631.2 million m$^3$ when the reservoir was completed in 1973. Operated by the SRWRO, its purposes include agricultural and public water supply, flood control, and hydropower generation. The annual inflow volume of the reservoir is around 120 million m$^3$ and the annual inflowing sediment volume is estimated as 5.6 million m$^3$ by the SRWRO.

Located 6 km downriver of the Tsengwen Reservoir, the East Weir diverts the releases from the Tsengwen Reservoir to the Wushan Hill Tunnel at a conveyance capacity of 56 m$^3$/s. The water is conveyed 3.3 Km to the West Weir on the Guantien Creek and then flows into the Wushanto Reservoir.



The Wushanto Reservoir is situated to the southwest of the Tsengwen Reservoir in the upper section of Guantien Creek, a tributary of Tsengwen River. The watershed area of the Wushanto Reservoir is only 60 km², which renders it conceptually off-stream. In 2011, its effective capacity was measured at 79.82 million m³. The Chianan Irrigation Association

manages the Wushanto Reservoir in coordination with the release from Tsengwen Reservoir, supplying irrigation water to over 70 thousand hectares of farmland in the Chianan Plain and providing the public and industrial water to the Tainan City and a portion of Chiayi City and Chiayi County. The joint operating rule curves for the Tsengwen and Wushanto Reservoirs are presented in Fig. 2 of Subsection 2.2.1.

From the beginning of operations in April 1973 until October 2017, the effective capacity of the Tsengwen Reservoir was reduced from 631.2 million m³ to 453.7 million m³. A major cause was Typhoon Morakot in 2009, which brought record-breaking rainfall to the reservoir watershed. The flood inflow of Tsengwen Reservoir peaked at 11,729 m³/s, which is only slightly below the peak of its probable maximum flood as 12,430 m³/s. Measurements at

the end of 2009 indicated that the sedimentation of Tsengwen reservoir had increased by a massive 91.08 million m³ that year, which is 19.7 times that of the average annual sedimentation between 1973 and 2008. In response to the substantial increase in sedimentation, the SRWRO improved the PRO and constructed a new desitling tunnel near the dam. The improvements of PRO include changing the original Howell-Bunger valve to a

jet flow gate and increasing the releasing capacity to 177 m³/s. The design capacity and minimum operating WSL of the desilting tunnel are 1,070 m³/s and 210 El.m respectively. At present, the elevation of bed in front of the dam of Tsengwen Reservoir have been raised beyond 171 El. m. This level is higher than the bottom of the inlets of the PRO at the 153.37 El. m, which allows empty flushing through the PRO.

The process of deriving the optimal empty flushing strategy starts from determining the

feasible period for implementing empty flushing to the Tsegnwen Reservoir, based on the criteria mentioned in subsections 2.1 and 2.2. The storage balancing curves for these two reservoirs are then modified before and during the evaluated feasible period to create favorable conditions to initiate empty flushing. Different storage thresholds for activation and

5 termination of an empty flushing operation are then tested to preliminarily assess the trade-off between desiltation and water supply. These storage thresholds are calibrated to maximize the desilting volume without inducing intolerable water shortage. Finally, numerical simulation and validation analysis are performed to verify the effectiveness of the derived strategy. Details of this process and the results are presented and discussed in the following section.

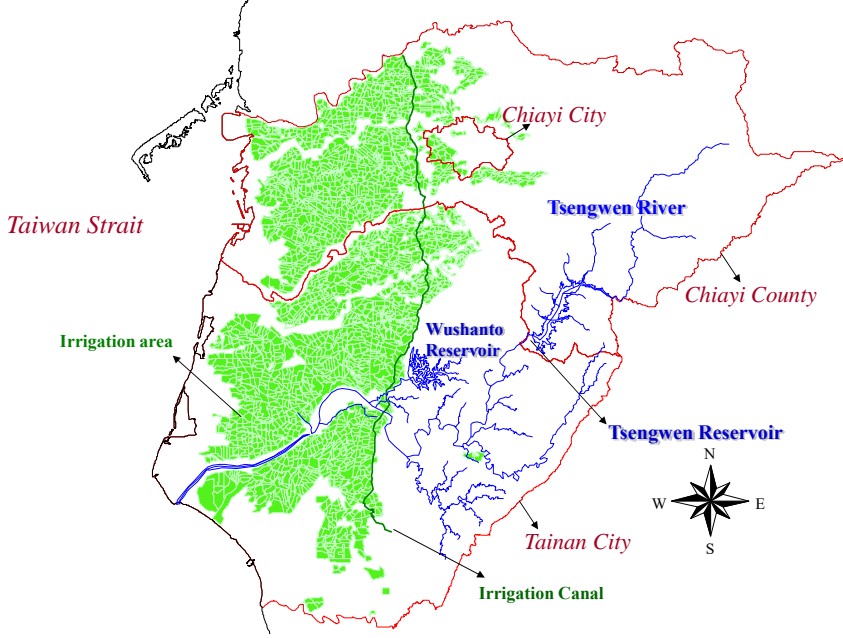

**Fig. 4 The map of Tsengwen and Wushanto Reservoirs**

## 3. Result and discussion

### 3.1 Determination of the feasible period for empty flushing

Fig. 5 illustrates both water demand of this system and average inflow to the Tsengwen Reservoir in ten-day increments over a year. As can be seen, inflow to the reservoir generally begins increasing between late May and early June, as precipitation rises during the beginning of the wet season. This is also the period in which the irrigational water demand, which constitutes the majority of total demands, is lower. The first semiannual rice crop is harvested, and the second semiannual irrigation just begins. As shown in Fig. 2, between May 11 and June 30, the lower limit of the operating rule curves is below the effective capacity of the Wushanto Reservoir. Even if the Tsengwen Reservoir is empty, as long as the Wushanto Reservoir is full, the total storage of the system would still exceed the lower limit of the operating rule curves, such that the demand for water could be satisfied. All of these characteristics indicate that the meteorological and operating conditions during May and June are favorable for empty flushing of Tsengwen Reservoir.

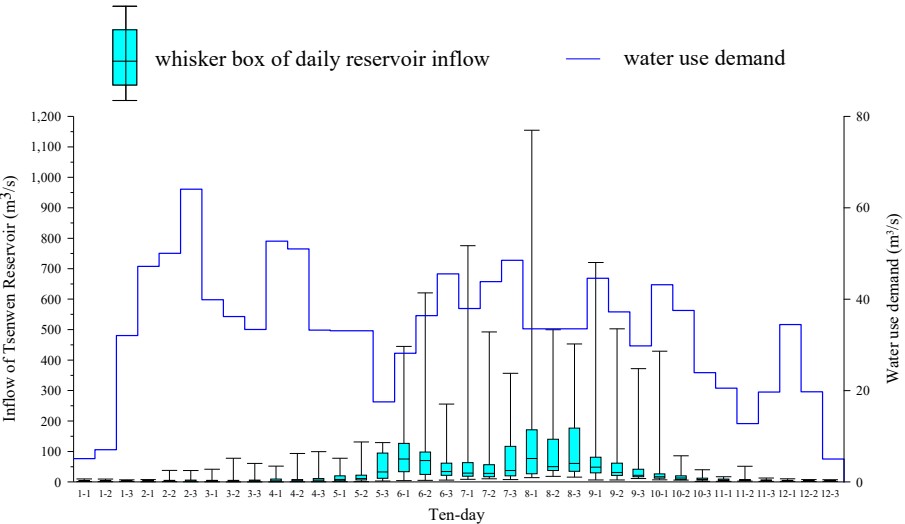

**Fig. 5 Demand and inflow patterns of Tsengwen Reservoir during ten-day periods throughout the year**



To validate the above assertion, sequential water supply simulation is performed using the daily inflow records of the reservoirs from 1975 to 2009 and the joint operating rules as described in subsection 2.2.1 in the absence of empty flushing operations. Fig. 6 illustrates the network of the water resources system. The simulated results provide a basis for calculating

5   the probability that the storage in the Tsengwen Reservoir drops below 20 million m³ for preparing empty flushing timely in a given month while the Wushanto Reservoir storage simultaneously exceeds the lower limit of the operating rule curves. The results are displayed in the "Balancing curves I" rows of Table 2. The results show that in May, there is a 52 % probability that the storage of the Tsengwen Reservoir will drop below 20 million m³ and an 8

10   % probability that the Tsengwen Reservoir storage drops below 20 million m³ while the Wushanto Reservoir storage simultaneously exceeds the lower limit. In June, the two probabilities are 31 % and 14 %, respectively. These two months present the highest probabilities during a year. The respective storage levels of the Tsengwen and Wushanto Reservoirs each satisfy the abovementioned conditions only between May 11 and June 20,

15   which is thus selected as the feasible period for empty flushing in the Tsengwen Reservoir.





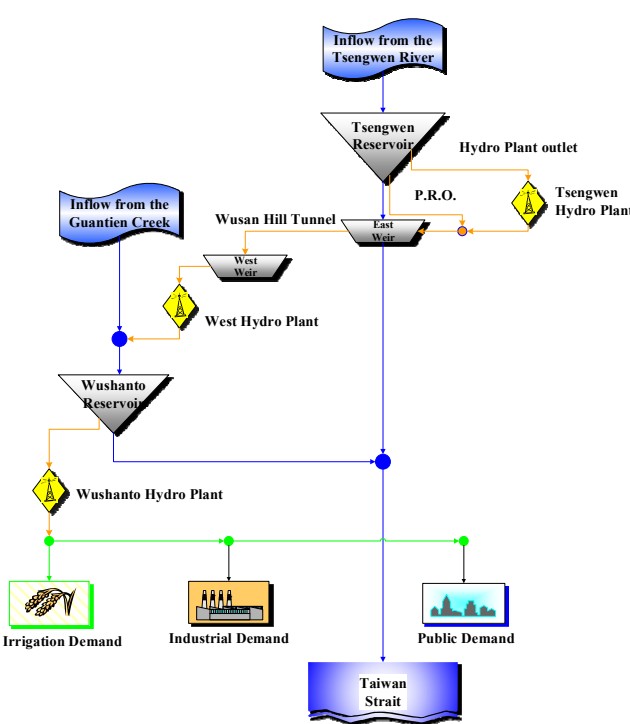

**Fig. 6 Network of the joint operating system of Tsengwen and Wushanto Reservoirs**

### 3.2 Schemes for the modification of storage balancing curves

5    The original storage balancing curves of the case study system such as Fig. 3 were

designed to ensure efficient utilization of water resources and adequate supply, without any

consideration of empty flushing. It was derived initially based on field operating experiences

and revised through trial and error process. To serve this study's purpose to maximize

opportunities for empty flushing of Tsengwen Reservoir, the balancing curves are modified to

10   preserve as much storage in the Wushanto Reservoir as possible by satisfying demands first

with Tsengwen Reservoir storage. Fig. 7 depicts the modified balancing curves. Based on the

simulation conditions given in subsection 3.1, three additional simulations are conducted in

which the modified storage balancing curves are applied during (1) May 1 to June 20, (2)

April 1 to June 20, and (3) March 1 to June 20. The simulated results are summarized in





Table 2, which include the monthly probabilities of the storage in the Tsengwen Reservoir dropping below 20 million $m^3$, the conditional probabilities of the storage in the Tsengwen Reservoir dropping below 20 million $m^3$ given that the storage in the Wushanto Reservoir being higher than the lower limit, and the monthly water shortage ratio. The results

demonstrate that the probability of favorable storage distribution for empty flushing during May and June can be effectively elevated by modifying storage balancing curves in April. The trade-off of creating favorable conditions for empty flushing is that preserving the storage of Wushanto Reservoir before and during empty flushing might cause unnecessary spillage while the reservoir is full and the inflow from Guantien Creek cannot be stored. Also,

allocating storage from the Twengwen Reservoir to Wushanto Reservoir through the Wushan Hill Tunnel will induce more transmitting loss of water. Nonetheless, the water shortage ratios generated by the modified balancing curves are no more than 0.01 higher than those from the original balancing curves, which means that the modification has only a trivial impact on the efficiency of water resources utilization.

The results also indicate that the average water shortage ratio during the wet season drops considerably after July. This is because the first typhoon of the wet season generally occurs in July or early August, bringing substantial inflow to the reservoirs. Thus in the following evaluation of empty flushing strategies, the water shortage scenarios through the end of July are selected to represent the impact of empty flushing on short-term water supply.

For the long-term impact, the shortage ratio during the next dry season, from January to May of the next year, are considered.



**Table 2 Monthly probabilities of Tsengwen Reservoir storage dropping below 20 million m³ under various strategies of storage allocation**

| Month | Jan | Feb | Mar | Apr | May | June | Jul | Aug | Sep | Oct | Nov | Dec |
|---|---|---|---|---|---|---|---|---|---|---|---|---|
| Index / Strategy | Monthly probabilities of Tsengwen Reservoir storage dropping below 20 million m³ | | | | | | | | | | | |
| Balancing curves I | 0.01 | 0.04 | 0.13 | 0.13 | 0.52 | 0.31 | 0.12 | 0.02 | 0.00 | 0.00 | 0.00 | 0.00 |
| Balancing curves II | 0.03 | 0.04 | 0.13 | 0.13 | 0.64 | 0.33 | 0.13 | 0.03 | 0.00 | 0.00 | 0.00 | 0.01 |
| Balancing curves III | 0.03 | 0.04 | 0.13 | 0.34 | 0.78 | 0.33 | 0.13 | 0.02 | 0.00 | 0.00 | 0.00 | 0.01 |
| Balancing curves IV | 0.03 | 0.04 | 0.13 | 0.45 | 0.78 | 0.33 | 0.13 | 0.02 | 0.00 | 0.00 | 0.00 | 0.01 |
| Index / Strategy | Monthly probabilities of storage in Tsengwen Reservoir dropping below 20 million m³ with storage in Wushanto Reservoir exceeding the lower limit | | | | | | | | | | | |
| Balancing curves I | 0.00 | 0.00 | 0.00 | 0.00 | 0.08 | 0.14 | 0.00 | 0.00 | 0.00 | 0.00 | 0.00 | 0.00 |
| Balancing curves II | 0.00 | 0.00 | 0.00 | 0.00 | 0.11 | 0.16 | 0.00 | 0.00 | 0.00 | 0.00 | 0.00 | 0.00 |
| Balancing curves III | 0.00 | 0.00 | 0.00 | 0.00 | 0.14 | 0.16 | 0.00 | 0.00 | 0.00 | 0.00 | 0.00 | 0.00 |
| Balancing curves IV | 0.00 | 0.00 | 0.00 | 0.01 | 0.14 | 0.16 | 0.00 | 0.00 | 0.00 | 0.00 | 0.00 | 0.00 |
| Index / Strategy | Average monthly water shortage ratio | | | | | | | | | | | |
| Balancing curves I | 0.10 | 0.17 | 0.23 | 0.29 | 0.22 | 0.07 | 0.10 | 0.06 | 0.06 | 0.09 | 0.09 | 0.11 |
| Balancing curves II | 0.11 | 0.18 | 0.24 | 0.30 | 0.23 | 0.07 | 0.10 | 0.06 | 0.07 | 0.09 | 0.09 | 0.12 |
| Balancing curves III | 0.11 | 0.18 | 0.24 | 0.31 | 0.23 | 0.07 | 0.10 | 0.06 | 0.07 | 0.09 | 0.09 | 0.12 |
| Balancing curves IV | 0.11 | 0.18 | 0.24 | 0.31 | 0.23 | 0.07 | 0.10 | 0.06 | 0.07 | 0.09 | 0.09 | 0.12 |

Balancing curves I: the original storage balancing curves as shown in Fig. 3.
Balancing curves II: adopting the modified curves as in Fig. 7 from May 1 to June 20.
Balancing curves III: adopting the modified curves as in Fig. 7 from April 1 to June 20.
Balancing curves IV: adopting the modified curves as in Fig. 7 from March 1 to June 20.

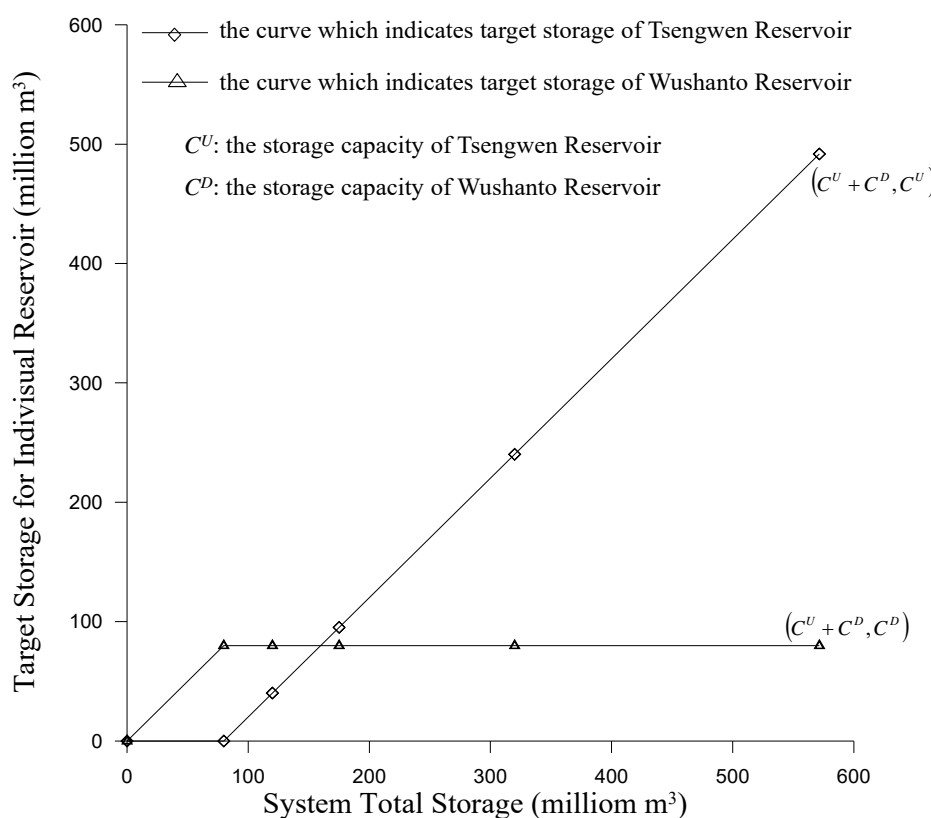

**Fig. 7 Modified storage balancing curves using Tsengwen Reservoir as primary**

**source to satisfy demand for water**

### 3.3 Preliminary simulations and assessment of empty flushing strategies

The thresholds to activate and terminate an empty flushing operation, i.e. $T^U$ for the

Tsengwen Reservoir, $T^D$ and $T^d$ for the Wushanto Reservoir, and $T^V$ for the maximum

accumulative releasing volume are regarded as parameters to be optimized. Except $T^V$, the

10  other parameters are allowed to vary during different ten-day periods from May 11 to June 20

in order to promote the performance of desilting and backup water supply. Before actually

optimizing these parameters, preliminary simulations are performed with constant storage





thresholds throughout May 11 to June 20. This process facilitates determination of a good initial solution as well as a basis for comparison to measure the effects of optimization. The preliminary simulations consider seven different $T^U$ values, ranging from 0 to 60 million m$^3$ with a constant interval of 10 million m$^3$, for the Tsengwen Reservoir. Six values (including

55, 60, 65, 70, 75 and 79 million m$^3$) for $T^D$ and nine values (including 30 to 70 million m$^3$ with a constant interval as 5 million m$^3$) for $T^d$ are considered. For the preliminary simulations, the value $T^V$ is set as an extremely high value, thus imposing no constraint on the water releasing volume of empty flushing as well as the long-term water supply. All these values contribute to a total of 308 combinations of empty flushing strategies in which $T^d$ is

less than $T^D$.

To estimate the volume of flushed sediments, measurements of sediment concentration from the PRO release of the Tsengwen Reservoir are used to establish the relationship between the flushing coefficient $\psi$ and the WSL of the reservoir, as shown in Fig. 8. The available concentration data with relatively lower WSL of reservoir were measured in early

wet seasons in 2004 and 2011 respectively. It reveals that during the regular periods of 2011 while reservoir WSL constantly exceeded El. 190 m, $\psi$ approached a fixed value of 1.0. The value of $\psi$ was elevated to a peak of 145 during a 5-days episode of heavy rainfall, which is the only flood in 2011. For this event of high WSL, the sediments vented by PRO were more probably from the turbidity current by the sediment-laden inflow, rather than being flushed

from existed depositions. Exact data from flushing does not exist since the reservoir has never been gone through such operations. The operators have constantly maintained the reservoir at high WSL to ensure stable water supply. The only experience for the reservoir drawing nearly to empty was in the beginning of wet seasons in 2004 before the Typhoon Mindulle invaded, during which the reservoir did not release water at all. Fig. 8 also includes the data of the





reservoir WSL, inflowing discharge and measured sediment concentration in front of the dam in a few hours of the rising periods of Typhoon-Mindulle-flood, during which the WSL had yet been raised beyond El. 190 m. It shows the corresponding value of $\psi$ ranges from 20 to 160, surrounding the value of 60 suggested by Atkinson (1996) for cases with limited-

capacity bottom outlets. Atkinson (1996) also suggested that when the water depth of a reservoir exceeds 30 % its maximum depth, the flushing efficiency will decrease significantly. The 30 % depth of the Tsengwen Reservoir is approximately at the elevation of 185 El. m, which corresponds to an impoundment of 16.6 million m$^3$. To prevent overestimating the effectiveness of empty flushing, this study assumes that if a flood raises the WSL of the

Tsengwen Reservoir to exceed 185 El. m, then the flushed sediment volume from the PRO is set to be 0, otherwise the $\psi$ is set as 60. In addition, the assumption of uniform flow condition during empty flushing allows the use of thalweg slope, which is 0.0032 according to the measurement, to represent the energy slope as required in Eq. (1). Then, according to the simulated PRO release during the empty flushing operation, the flushed sediment

discharge as well as the desilting volume during the simulating time horizon can be estimated using Eq. (1).

It should be noted that similar setting of $\psi$ has been applied and validated in the Akungten Reservoir in an adjacent basin of southern Taiwan (SWRO, 2015). This reservoir annually undergoes empty flushing between 1$^{st}$ May to 10$^{th}$ September. The estimated

volumes of flushed sediments during several floods appears to be in the same order to the measurements as shown by Table 3. Substituting the above setting of $\psi$ and a full-capacity discharge of PRO of Tswengwen Reservoir into Eq. (1) will lead to a volumetric sediment concentration as 39,928 ppm. Another adjacent Jensanpei Reservoir in southern Taiwan recorded the volumetric sediment concentration from its historical empty flushing operations





ranging between 54,014 to 446,182 ppm, and averaged at 105,478 ppm (Water Resources

Planning Institute, 2010). This shows the conservativeness of the adopted setting of $\psi$ in the

current study.

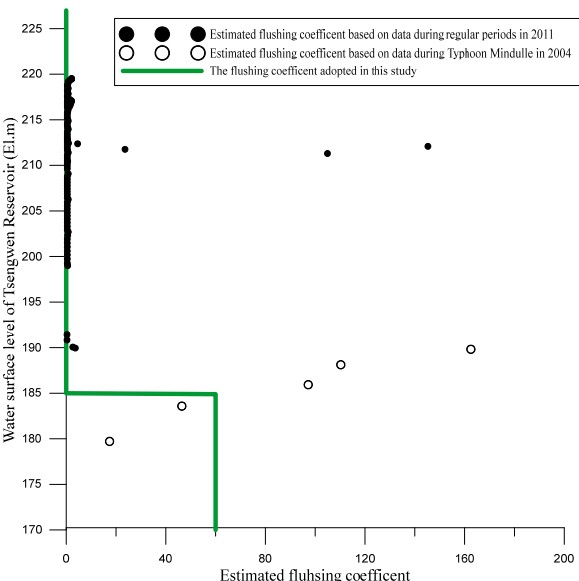

**Fig. 8 Relationship between the flushing coefficient and WSL of Tsengwen Reservoir**

**Table 3 The operating records of Akungten Reservoir during its empty flushing periods (SWRO, 2015)**

| Typhoon event | Duration | Total rainfall (mm) | Peak inflow (m³/s) | Total desilting mass (thousand ton) | |
|---|---|---|---|---|---|
| | | | | Measurement | Estimation |
| Morakot | 2009/08/06~08/11 | 836.5 | 303.2 | 67.0 | 51.0 |
| Fanapi | 2010/09/18~09/21 | 589.0 | 673.9 | 16.0 | 20.6 |
| Nanmadol | 2011/08/27~08/31 | 353.0 | 95.4 | 32.0 | 41.4 |
| Torrential Rain | 2012/06/09~06/16 | 593.0 | 248.8 | 48.0 | 47.4 |
| Talim | 2012/06/18~06/25 | 348.0 | 164.7 | 14.0 | 5.2 |
| Trami | 2013/08/21~08/24 | 225.0 | 127.0 | 6.0 | 14.3 |
| Kong-Rey | 2013/08/29~09/03 | 546.5 | 216.7 | 3.0 | 15.9 |
| | | | Average | 26.6 | 28.0 |





**Table 4 The operating records of Jensanpei Reservoir during empty flushing operations**

**from 1957 to 1980 (Water Resources Planning Institute, 2010)**

| Duration | Total rainfall during flushing (mm) | Volume of reservoir release (m³) | Volume of flushed sediment (m³) | Volumetric concentration (ppm) |
|---|---|---|---|---|
| 1957/05/13~07/14 | 1064.0 | 7,611,755 | 672,900 | 88,403 |
| 1958/05/23~07/14 | 719.4 | 1,919,864 | 521,224 | 271,490 |
| 1959/05/31~07/20 | 761.5 | 3,554,264 | 295,529 | 83,148 |
| 1960/05/14~08/01 | 660.6 | 3,108,825 | 528,470 | 169,990 |
| 1961/05/18~08/07 | 520.4 | 2,609,121 | 503,868 | 193,118 |
| 1962/06/07~08/02 | 833.2 | 4,068,302 | 434,615 | 106,830 |
| 1963/06/03~07/18 | 553.6 | 4,822,534 | 317,950 | 65,930 |
| 1970/05/31~07/31 | 266.6 | 1,153,827 | 103,618 | 89,804 |
| 1971/06/03~07/25 | 659.6 | 3,956,783 | 251,635 | 63,596 |
| 1972/06/05~07/12 | 682.5 | 5,703,971 | 308,096 | 54,014 |
| 1973/06/02~07/17 | 404.3 | 2,811,910 | 287,515 | 102,249 |
| 1974/06/14~07/17 | 513.9 | 3,383,116 | 423,299 | 125,121 |
| 1975/06/05~07/02 | 771.6 | 5,252,552 | 443,908 | 84,513 |
| 1976/06/11~07/15 | 527.0 | 4,375,959 | 317,441 | 72,542 |
| 1977/05/29~07/01 | 1134.0 | 9,436,599 | 839,701 | 88,983 |
| 1978/05/19~07/30 | 362.5 | 620,895 | 123,390 | 198,729 |
| 1979/05/28~06/28 | 340.0 | 950,487 | 408,755 | 430,048 |
| 1980/06/21~07/30 | 117.5 | 323,088 | 144,156 | 446,182 |
| average | 605.1 | 3,647,992 | 384,782 | 105,478 |

To assess the impact on short-term water supply following empty flushing of the preliminary simulations, the ratio and increments of water shortages during the remaining periods of June and during the entire July in each simulated year are calculated. The maximum short-term monthly water shortage ratio is then calculated according to the following equations:

$$d_{\max,0}^{R} = \max_{n=1,\dots,n^{y}} (d_{n,0}^{R} | d_{n,0}^{I} > 0) \tag{9}$$

$$d_{\max,1}^{R} = \max_{n=1,\dots,n^{y}} (d_{n,1}^{R} | d_{n,1}^{I} > 0) \tag{10}$$

$$d_{\max}^{R} = \max(d_{\max,0}^{R}, \ d_{\max,1}^{R}) \tag{11}$$

where $d_{n,0}^{R}$ and $d_{n,0}^{I}$ denote the ratio and increments of water shortage between the end of the last empty flushing operation to the end of June in the $n^{th}$ simulated year, while $d_{n,1}^{R}$ and $d_{n,1}^{I}$ are the ratio and increments of shortage throughout July in the $n^{\text{th}}$ year; $d_{max,0}^{R}$ and $d_{max,1}^{R}$ represent the maximum water shortage ratio during the period between the completion of the last flushing operation until the end of June and during July, respectively. The primary causes of short-term water shortage included the absence of heavy rainfall during May and June and a delayed arrival of the first typhoon in July. These conditions lead to insufficient inflow and reservoir storage during June and July, which necessitate water rationing according to the joint operating rule curves.

According to the above conditions, simulations of the 308 combinations are performed using the original storage balancing curves. The resulting average annual desilting volume and maximum short-term monthly water shortage ratio induced by empty flushing are then calculated. The results are presented in Fig. 9. The simulations are then repeated by applying the modified storage balancing curves in Fig. 7 to the period between April and June, the results of which are displayed in Fig. 10. A comparison of Figs. 9 and 10 shows that the modified storage balancing curves effectively enhance the effectiveness of desilting. For instance, strategies with $d_{max}^{R}$ between 0.17 and 0.23 correspond to a maximum annual desilting volume of 0.06 million m$^3$/year in Fig. 9, whereas the same strategies in Fig. 10 result in an increase of desilting volume reaching 0.54 million m$^3$/year.





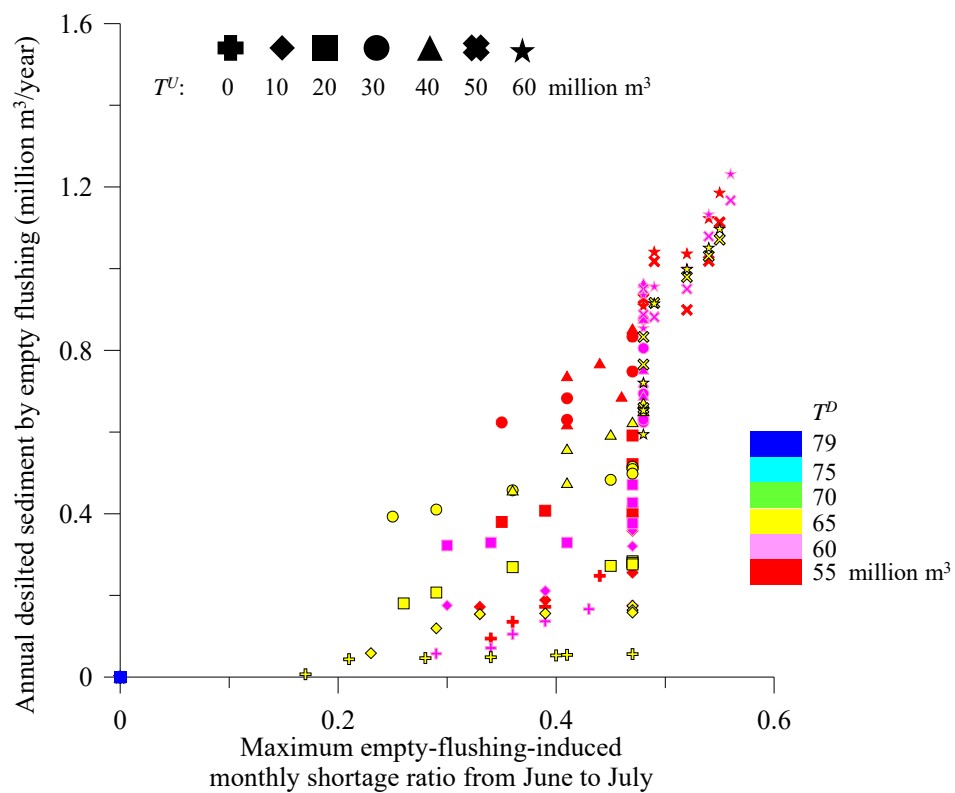

Fig. 9 Simulation results of various empty flushing strategies using the original storage

balancing curves

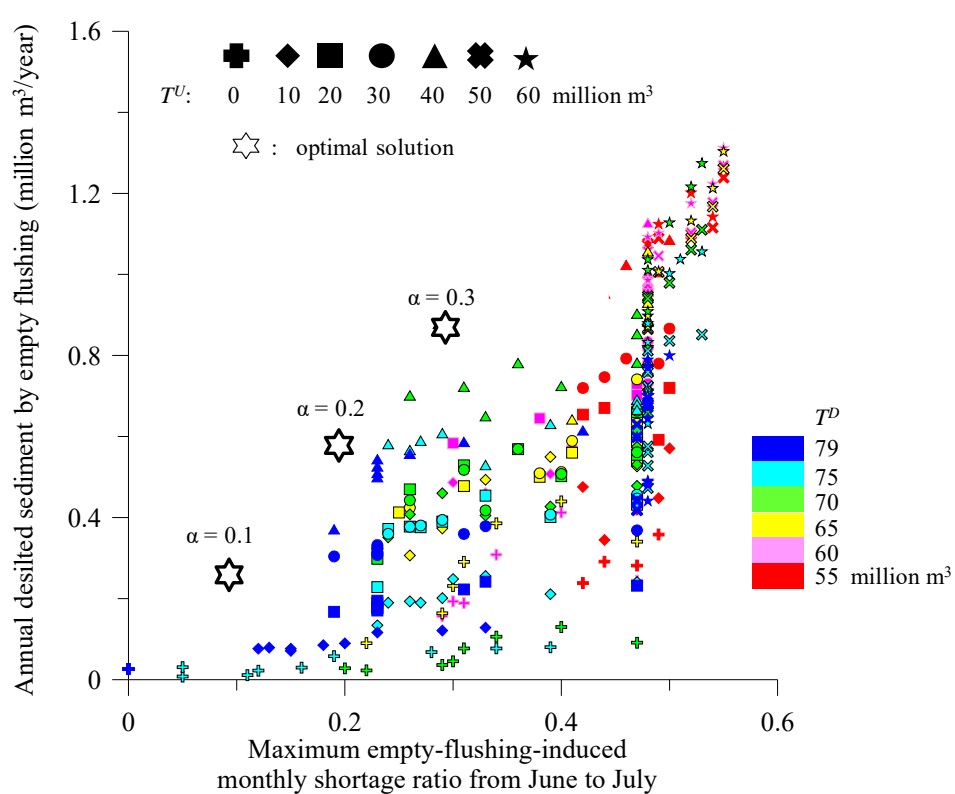

**Fig. 10 Simulation results of various empty flushing strategies using modified storage**

**balancing curves from April to June**

### 3.4 Optimization of empty flushing strategies

5       According to the shortage indices adopted in subsection 3.3, the constraint of Eq. (7) is

divided as follows to enable greater precision in control over short-term water shortages

induced by empty flushing:

$$d_{\max,0}^{R} = \max_{n=1,\ldots n^{y}}(d_{n,0}^{R}\big|d_{n,0}^{I} > 0) \leq \alpha \tag{12}$$

$$d_{\max,1}^{R} = \max_{n=1,\ldots,n^{y}}(d_{n,1}^{R}\big|d_{n,1}^{I} > 0) \leq \alpha \tag{13}$$



For the constraint of Eq. (8), water shortage in the next dry season is calculated from the simulated result from January to May in the next year following empty flushing. Coupling the established simulation model which simulates the water supply and empty flushing process from 1975 to 2009 with the optimization algorithm leads to a preliminary optimal

5   solution of Eqs. (6), (8), (12) and (13) under specific values of $\alpha$ and $\beta$. Three sets of $\alpha$ and $\beta$ are tested, which are (0.1, 0.25), (0.2, 0.30) and (0.3, 0.35) respectively. The corresponding optimal storage thresholds to activate and terminate an empty flushing operation are presented in Table 5. The average annual desilting volume and maximum monthly shortage ratio induced by empty flushing are also marked in Fig. 10.

10   Due to the frequency of drought in this system, the optimal strategy associated with $\alpha$=0.1 and $\beta$=0.25 is selected for further scrutiny. Table 6 displays the simulated events of empty flushing based on this calibrated strategy.





**Table 5 Optimal empty flushing strategies based on acceptable water shortage rates following the completion of empty flushing operations**

| Ten-day period | Tsengwen Reservoir flushing initiation condition TU (million m³) | | | | Wushanto Reservoir flushing initiation condition TD (million m³) | | | | Wushanto Reservoir flushing termination condition Td (million m³) | | | | The maximum allowable accumulative water release volume from Twengwen Reservoir (million m³) | Annual water shortage increment (million m³/year) | Annual desilting volume (million m³/year) |
| --- | --- | --- | --- | --- | --- | --- | --- | --- | --- | --- | --- | --- | --- | --- | --- |
| | 14th | 15th | 16th | 17th | 14th | 15th | 16th | 17th | 14th | 15th | 16th | 17th | | | |
| $\alpha = 0.1,\ \beta = 0.25$ | 0.2 | 40.0 | 40.0 | 0.0 | 75.0 | 70.0 | 68.5 | 79.8 | 73.9 | 51.0 | 60.0 | 76.0 | 76.0 | 11.28 | 0.25 |
| $\alpha = 0.2,\ \beta = 0.30$ | 60.0 | 40.0 | 44.0 | 30.0 | 75.0 | 61.0 | 67.5 | 79.8 | 55.0 | 40.0 | 44.0 | 60.0 | 132.0 | 14.74 | 0.59 |
| $\alpha = 0.3,\ \beta = 0.35$ | 80.0 | 40.0 | 40.0 | 30.0 | 75.0 | 57.0 | 67.5 | 79.8 | 51.1 | 40.0 | 39.2 | 60.0 | 199.0 | 17.56 | 0.88 |



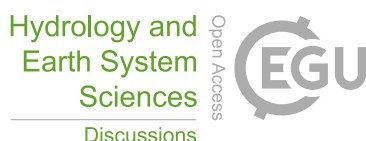

**Table 6 Simulated empty flushing events based on the optimal strategy with $\alpha = 0.1$, $\beta = 0.25$**

| Year | Duration | Inflow volume to T. Reservoir (M.m³) | Reservoir storage at initiation (M.m³) T. | W. | Reservoir storage at termination (M.m³) T. | W. | Water shortage until June increment (M.m³) | ratio (%) | Water shortage in July increment (M.m³) | ratio (%) | Water shortage ratio during the next dry season (%) w.o./e.f. | w./e.f. | Bottom release volume from T. Reservoir (M.m³) | De-silting volume (M.m³) | Ave. concentration (ppm) | Down-stream lateral flow during e.f. (M.m³) | Total Inflow to T. reservoir during the wet season (M.m³) | Spillage of T Reservoir in the wet season (M.m³) |
|---|---|---|---|---|---|---|---|---|---|---|---|---|---|---|---|---|---|---|
| 1979 | 6/10 | 25.03 | 11.60 | 69.10 | 21.06 | 69.07 | 0.00 | 0.00 | 0.00 | 0.00 | 17.0 | 17.0 | 11.12 | 0.40 | 36,291 | 7.67 | 1,109 | 155.80 |
| 1984 | 5/13~5/14 | 2.21 | 0.00 | 78.53 | 0.00 | 72.97 | 0.00 | 0.00 | 0.00 | 0.00 | 45.0 | 49.0 | 2.21 | 0.04 | 18,194 | 1.32 | 676 | -- |
|  | 5/21~5/31 | 121.55 | 0.00 | 70.26 | 42.68 | 62.45 |  |  |  |  |  |  | 65.27 | 1.26 | 19,344 | 30.51 |  |  |
| 1986 | 5/21~5/29 | 91.44 | 11.61 | 71.93 | 8.31 | 64.96 | 0.00 | 0.00 | 0.00 | 0.00 | 12.0 | 25.0 | 77.50 | 2.63 | 33,909 | 18.31 | 843 | -- |
| 1989 | 5/14~5/15 | 1.78 | 0.00 | 78.83 | 0.00 | 73.24 | 0.00 | 10.00 | 0.00 | 41.00 | 1.0 | 1.0 | 1.78 | 0.03 | 17,010 | 0.35 | 1,006 | 121.60 |
| 1997 | 6/07~6/10 | 31.99 | 5.58 | 71.08 | 0.00 | 64.45 | 0.00 | 0.00 | 0.00 | 0.00 | 5.0 | 5.0 | 33.25 | 1.11 | 33,381 | 2.84 | 941 | 23.51 |
| 2006 | 5/25~6/05 | 127.63 | 6.71 | 72.66 | 25.62 | 62.40 | 0.00 | 0.00 | 0.00 | 0.00 | 20.0 | 21.0 | 86.26 | 2.43 | 28,115 | 17.06 | 1,657 | 503.86 |
| 2008 | 6/09~6/10 | 7.59 | 28.72 | 68.54 | 51.22 | 64.41 | 0.00 | 0.00 | 8.71 | 7.00 | 1.0 | 2.0 | 22.28 | 0.81 | 36,311 | 1.22 | 1,930 | 858.88 |
| 2009 | 5/11~5/12 | 1.36 | 0.00 | 77.86 | 0.00 | 72.24 | 0.28 | 2.00 | 0.00 | 41.00 | 28.0 | 28.0 | 1.36 | 0.02 | 18,001 | 0.03 | 1,554 | 744.37 |
| 2010 | 6/04-6/08 | 11.61 | 21.02 | 69.74 | 0.00 | 58.70 | 0.00 | 0.00 | 7.53 | 18.00 | 2.0 | 4.0 | 23.85 | 0.74 | 30,992 | 0.53 | 927 | -- |
| 2013 | 5/11 | 1.47 | 0.00 | 75.56 | 0.00 | 72.81 | 0.00 | 0.00 | 0.00 | 0.00 | 20.0 | 20.0 | 1.47 | 0.03 | 19,767 | 0.63 | 1,347 | 402.24 |
|  | 5/21-5/31 | 64.37 | 19.91 | 70.71 | 0.00 | 58.72 |  |  |  |  |  |  | 66.59 | 2.12 | 31,906 | 6.52 |  |  |
| 2014 | 6/06 | 6.85 | 23.25 | 68.86 | 14.45 | 67.09 | 0.00 | 0.00 | 23.04 | 21.00 | 47.0 | 54.0 | 11.19 | 0.41 | 36,351 | 6.52 | 683 | -- |

1. T. and W. represent Twengwen and Wushanto Reservoirs respectively.

2. E.f. is the abbreviation of empty flushing.

3. The monthly shortage ratio in July following the empty flushing operations in 1989 and 2009 both reach 0.41. However, the corresponding shortage increments are both 0; therefore, they did not violate the constraints of Eqs. (12) and (13). For these events, the Tsengwen Reservoir is nearly empty and the Wushanto Reservoir is nearly full before the initiation of empty flushing operations. Thus, the empty flushing operations only consume the inflow of Tsengwen Reservoir during a 2 days period. These water consumption volumes are too insignificant to induce the subsequent water shortage seen in July. The primary reason for the subsequent shortage is the delayed arrival of the first typhoon in late July or early August, by which time the total storage falls below the critical limit of the joint operating rule curves and water rationing is applied. Following the arrival of the typhoons, however, the total reservoir storage exceeds the lower limit and even the upper limit of rule curves, thereby alleviating the water shortage.





### 3.5 Sensitivity analysis

The major uncertainty of the above analysis is the use of the empirical formula Eq. (1) to estimate the volume of flushed sediments. Due to the lack of field data, the flushing coefficient $\psi$ is directly assigned as 60, a most common and conservative value found in

literatures (Atkinson, 1996). To investigate how this uncertainty affects the optimization procedure, sensitivity analysis is performed by applying another two values of $\psi$ as 180 and 300, which are originated from IRTCES (1985). It turns out that the re-calibrated parameters by different values of $\psi$ remain unchanged. To demonstrate this characteristic, sensitivity simulation trials are run by expanding the storage thresholds to activate flushing, $T^U$ and $T^D$,

of the 15th ten-day from the optimal solution, with a discrete interval of 1 million m³. Fig. 13 depicts the contour maps of desilting volume and maximum monthly shortage ratio induced by empty flushing with axis as $T^U$ and $T^D$ for different values of $\psi$. It reveals that using a different $\psi$ value only leads to a linearly-varied value of the objective function, while the optimal solution is dictated by the water shortage constraint and free of $\psi$. This validates the

ability of the proposed approach to produce strategies which maximize the potential desilting performance under acceptable water shortage scenarios. Nonetheless, the precise magnitude of desilting volume still requires further investigation for a more accurate evaluation of the benefits of empty flushing. The next sub-section displays the result from numerical simulations on significant events of empty flushing in Table 6.





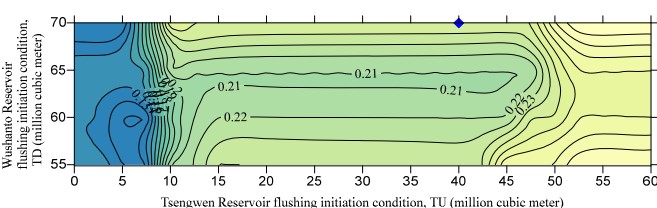

(a) Contour of the desilting volume for different $T^U$ and $T^D$ with $\psi$ = 60

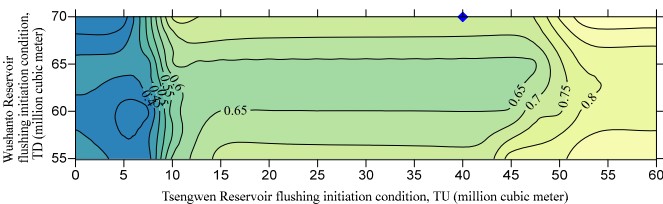

(b) Contour of the desilting volume for different $T^U$ and $T^D$ with $\psi$ = 180

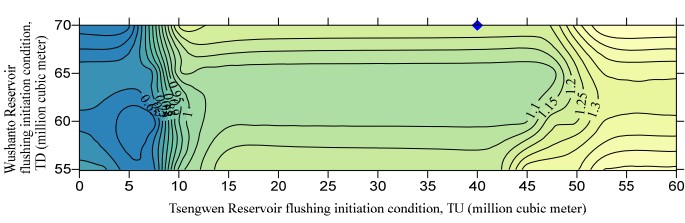

(c) Contour of the desilting volume for different $T^U$ and $T^D$ with $\psi$ = 300

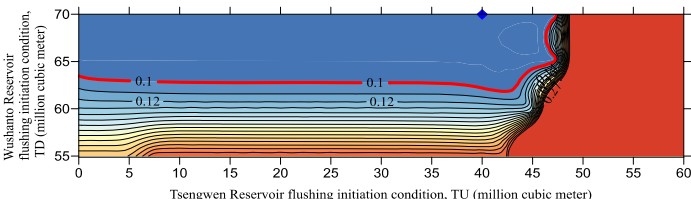

(d) Contour of the maximum monthly shortage ratio from June to July for different $T^U$ and $T^D$ with $\psi$ = 60, 180

10 or 300

**Fig. 13 Results of sensitivity analysis of flushing coefficient and threshold storages for**

**the 15th ten-day**





### 3.6 Numerical simulation of the empty flushing process

Flow in reservoir during empty flushing is relatively shallow and the effect of vertical motions is negligible. The 3D Navier-Stokes equations coupled with sediment transport formulas may be vertically averaged to obtain a set of depth averaged 2D equations. A

numerical model solving these equations, SRH-2D, is applied to validate the releasing sediment discharge under the condition of empty flushing. An extensive list of SRH-2D calibrations, verifications and applications had been carried out (Lai, 2008, 2010). The model can be downloaded from the website of the U.S. Bureau of Reclamation.

The flushing events in 1984, 1986, 1997 and 2006 listed in Table 6 are selected for

numerical investigation. The Dirichlet boundary conditions are given at upstream and downstream simulation grids. Sediment transport equation of Engelund-Hansen (1967) is adopted to simulate sediment discharge. The particle size of sediment deposited on the reservoir bottom is given from field survey. Fig. 14(a) shows the simulated bottom variation of the event in 2006. It shows that the erosion mechanism dominates the reservoir bed

changing due to low water level. A flushing channel is created from the inlet of sluiceway toward the midstream of the reservoir. The flushing process, including discharges of reservoir inflow and release, storage, sediment concentration of release and accumulative desilting volume during this event is also depicted in Fig. 14(b). The entire process consists of two stages of drawdown, empty and refill phases regarding reservoir storage. The release

concentration peaks around 50,000 mg/l while the reservoir is empty and the flushing channel is formed. The release with lower concentration during the two refill phases also facilitates alleviating the impacts on downstream environment. Figure 14(c) reveals that the simulated desilting volumes by SRH-2D agree well with the estimated values from Eq. (1). This validates the acceptability of the adopted empirical formula.

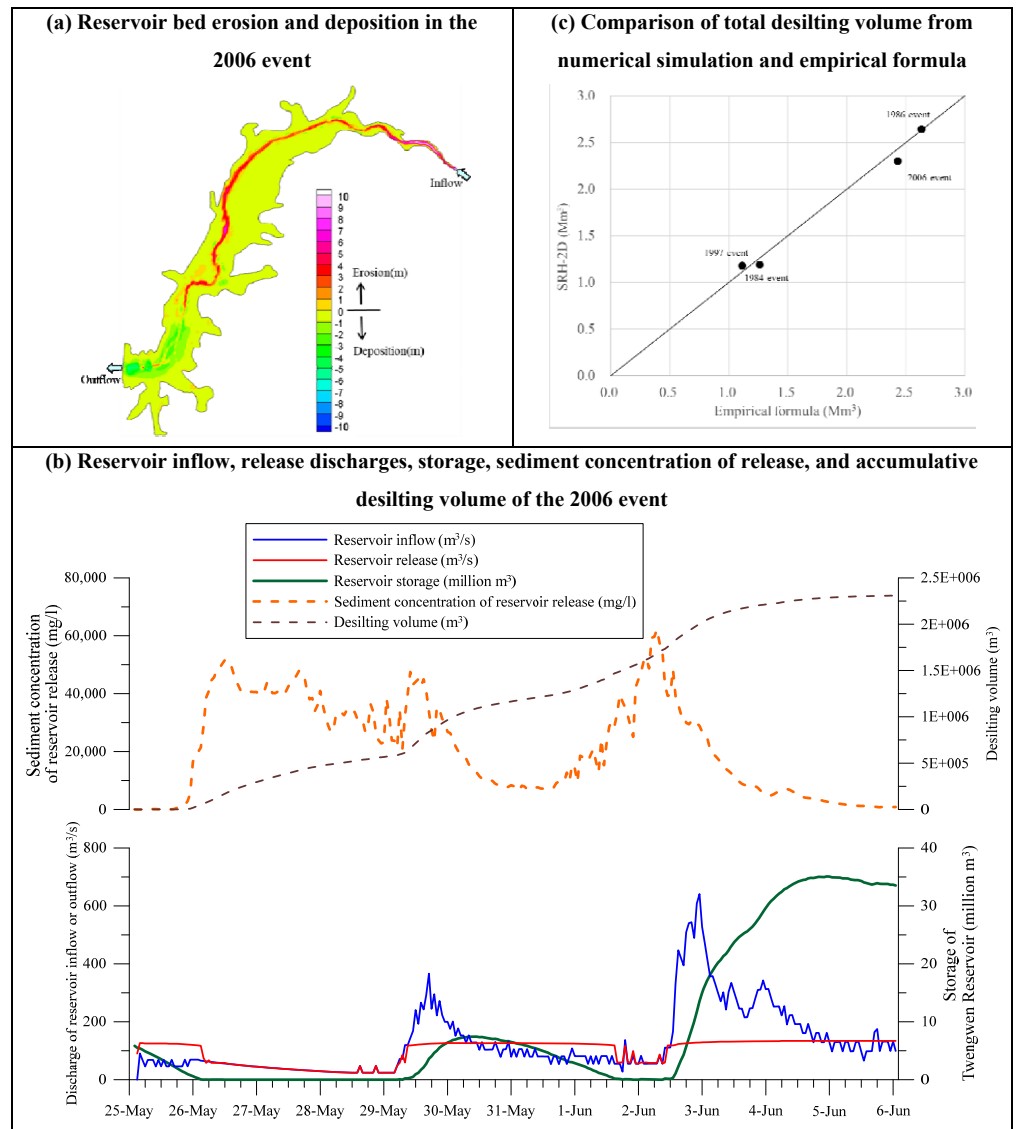

**Fig. 14 Simulation results of the SRH-2D**

## 3.7 Validation of the optimal strategies

The optimal strategies in Table 5 are derived according to the records of daily reservoir inflow between 1975 and 2009. Following this calibration period, the records through the end of 2017 are used to verify the effectiveness of the established strategy. The results indicate that four flushing operations could have been conducted during 2010 to 2017. The last four





rows in Table 6 summarize these events. Figs. 15 to 17, respectively, present the hydrographs of reservoir inflow and total system storage from May to December of these years. Fig. 18 shows the sedimentation progress of Tsengwen Reservoir from 1974 to 2017 with and without empty flushing. In addition to the case in 2013, which imposes no impact on the

following water supply, the empty flushing operations in the other validating years are discussed as below:

1. Following the initiation of empty flushing in early June of 2010, the monthly water shortage ratio during July is 0.18, which is higher than the 0.12 that would have been the case without empty flushing. The increased shortage ratio is induced by drawdown and

empty flushing, which cause the total storage to fall below the critical limit of rule curve earlier in July. Empty flushing thus necessitates a longer water rationing period. Nonetheless, torrential rains in late July elevate the storage to exceed the lower limit, thereby resolving the shortage crisis. The major impact of water shortage during this period is on the second semiannual irrigation operation, which requires large quantities of

water during July. One of the mitigation measures is to postpone the beginning irrigation schedule no later than August 10. For example, in June of 2004, the total storage in the Tsengwen and Wushanto Reservoirs fell below the critical limit, which delayed the second semiannual irrigation from the originally planned June 6 to July 17 when Typhoon Mindulle invaded and elevated the storage above the upper limit in one day of early July.

2. In 2014, over 40 % of the inflow of Twengwen Reservoir during wet season occurred in May and June. The unexpected lack of subsequent wet-season floods leads to failure on recovery of reservoir storage. The impact is alleviated through water rationing during the wet season to allow carrying out the second semiannual irrigation. For the dry season in 2015, the drought would inevitably lead to large scale suspension of the first semiannual

irrigation, whether empty flushing in the previous year is performed or not.

As for the impact on downstream environment, either the bottom release during empty flushing or the spillway release during the rest of the year in almost every event in Table 6 exceeds the required volume, 40 million m$^3$, to transport depositions downstream from the Twengwen Reservoir. The only two exceptions are the events of 1977 and 2010. In order to alleviate the impacts by empty flushing as well as to recover deposition area for dredging, the reservoir may need to deliberately release water before the end of wet season. This will increase the shortage situation in the next dry season by 40 million m$^3$, which is about 10% of the first semiannual irrigation demand. For these two years, this shortage increment can still be managed without inducing suspension of the following irrigation.





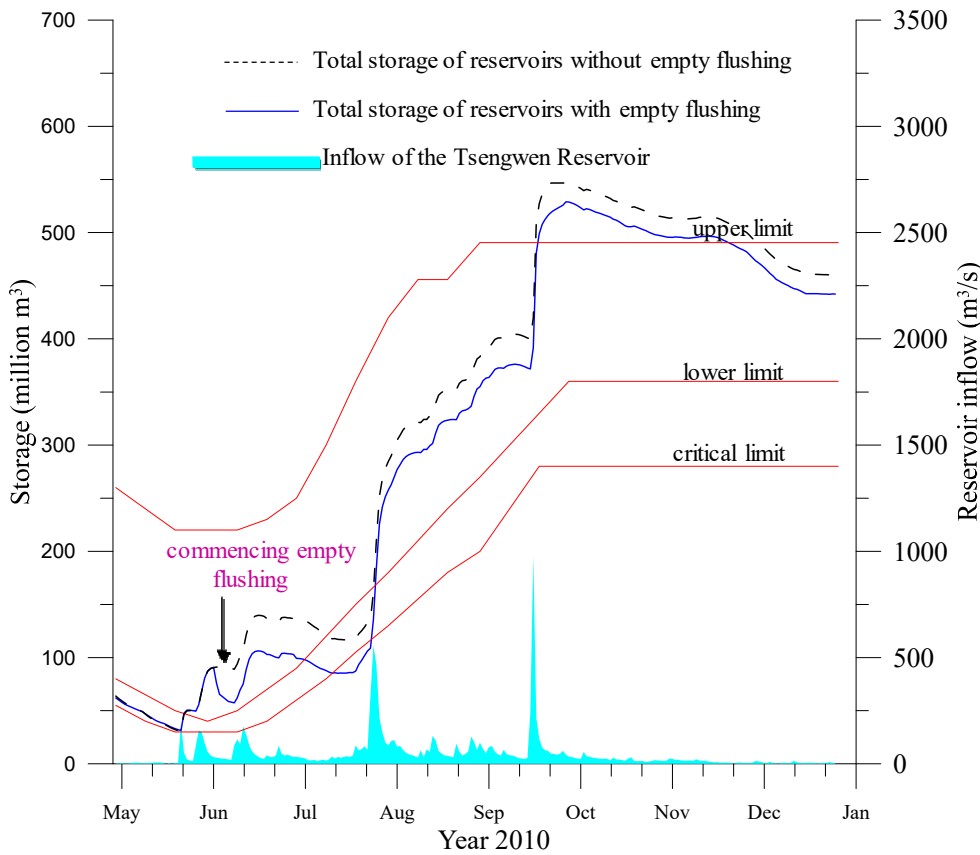

Fig. 15 Reservoir inflow and storage throughout 2010





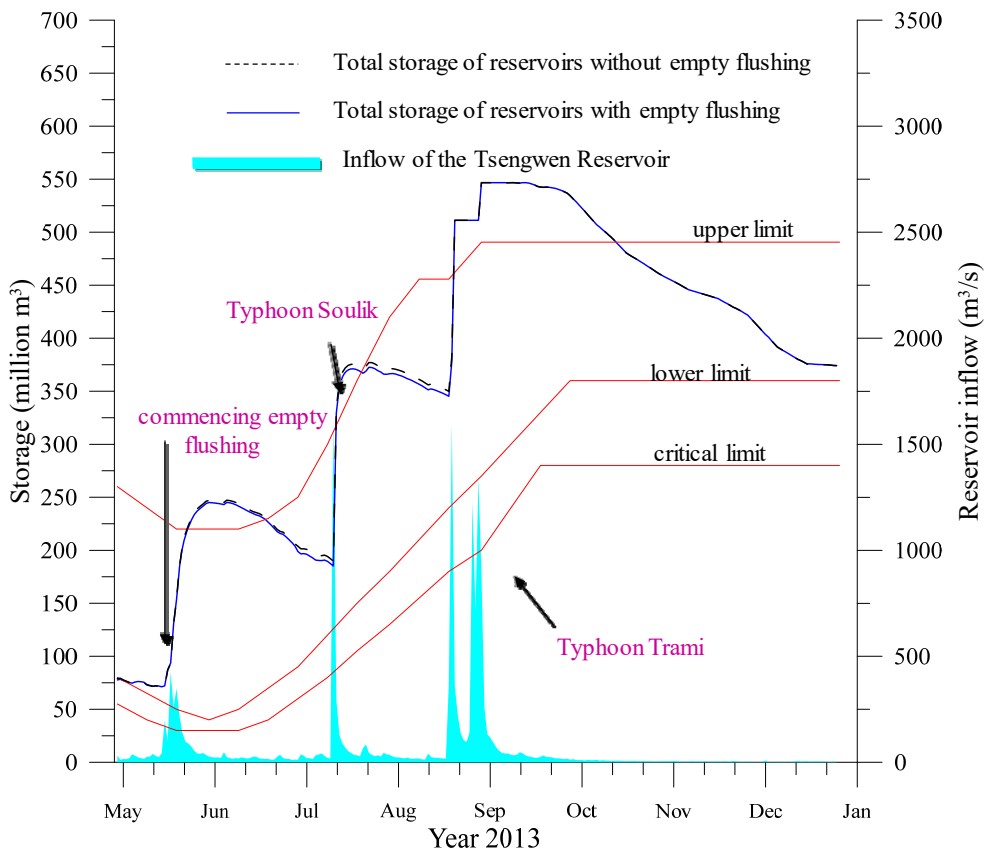

**Fig. 16 Reservoir inflow and storage throughout 2013**


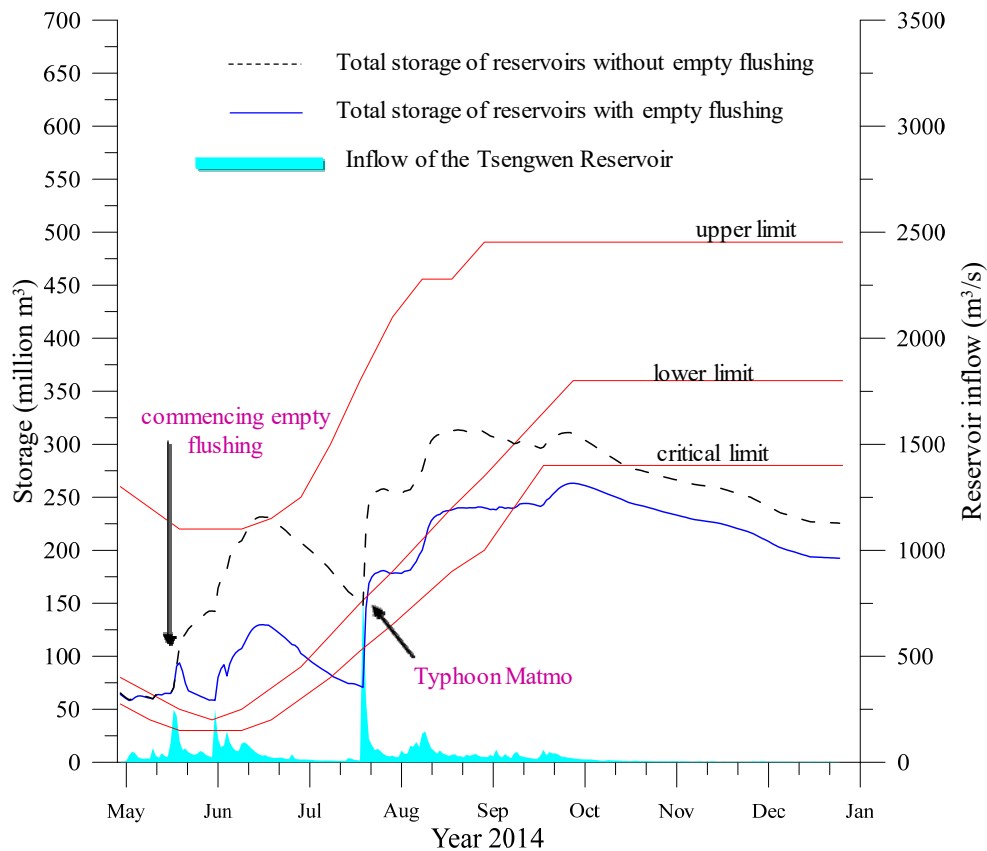

**Fig. 17 Reservoir inflow and storage throughout 2014**

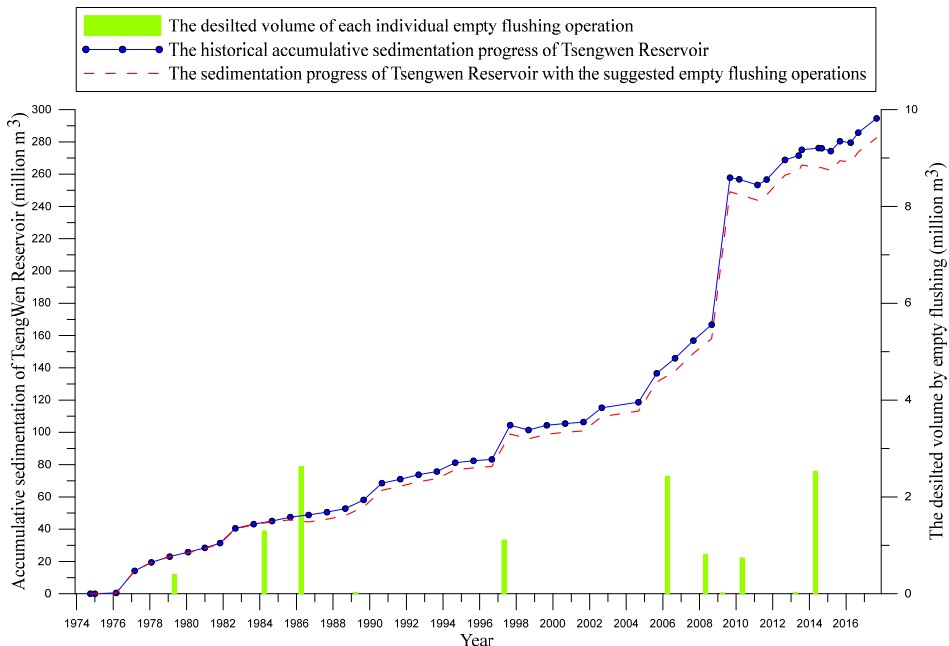

**Fig. 18 The sedimentation progress of Tsengwen Reservoir with and without empty flushing**

### 3.8 Potential extensions of the proposed method

Some suggestions regarding future extension of the current study is provided as below:

1. To incorporate short-term forecasting of reservoir inflow in determining the activation or termination of an empty flushing operation. If the forecast is reliable, a certain amount of storage can be kept in the primary reservoir and pre-emptied shortly before an expected flood. The flood flow will be effective for scouring the deposited sediments of the primary reservoir if the capacity of its bottom outlet is adequate to fully vent the inflow. The forecast uncertainty should be incorporated into the model predictive control framework or the decision analysis such as suggested by Chou and Wu (2013) to avoid inducing intolerable water shortage in case the flood does not occur after empting the reservoir.





2. The case study specifically discusses a system with an upstream online reservoir requiring empty flushing and a downstream offline reservoir which provides backup water supply. Implementations for other systems with different schematics may require more complex optimization formulations with additional parameters. For example, if an additional

reservoir is available upstream from the primary reservoir, its storage could also serve to generate flushing discharge. The flushing operation may start from emptying the downstream reservoir, and then drawing down the storage of the upstream reservoir and allowing the drawdown release to scour and pass through the downstream reservoir. The timing to start and terminate the joint flushing operation may be guided by the joint

operating rule curves. The joint rule curves can also be included as parameters to be optimized to promote the performance of empty flushing without inducing significant water shortage. Similar study can be found in Chou and Wu (2017).

3. The objective function can be updated to maximize the net benefits of enhanced desilting by empty flushing from more frequent transfer of agricultural water to public supply. This

may require a more accurate estimation of the desilting volume. Numerical modeling may be either employed to simulate the flushing process, estimate the desilting volume and validate the setting of $\psi$ , or directly integrated into the optimization framework. The transportation of flushed sediments in the downstream river could also be simulated to more comprehensively evaluate of the impacts of empty flushing on environment.

## 4. Conclusions

This study aims to optimize the performance of empty flushing of one primary reservoir within a multi-reservoir system. Prior to empty flushing, the total available storage in a system is allocated from the primary reservoir to the others in order to create favorable initial



conditions and prepare backup water to be supplied during empty flushing. The activation and termination conditions of an empty flushing operation are determined according to whether storage in the primary and auxiliary reservoirs satisfies applicable thresholds. Optimization analysis calibrates these storage thresholds to maximize the desilting volume without

inducing intolerable water shortage. The case study of the water resources system of the Tsengwen and Wushanto Reservoirs of southern Taiwan verifies the effectiveness of the derived optimal strategy.

Integrating reservoir desilting considerations with water supply operation creates more facets into the multi-objective water resources management. In addition to irrigation,

municipal, industrial and hydropower purposes, the competition of water extends to include sediment flushing, sluicing, vacating previous dredged and deposited sediments, and alleviating their impacts on downstream environment. The high risk of water shortage in the case study area currently dictates the operating objective to solely focus on reliable water supply. This restricts the feasibility of not only empty flushing, but any other operations may

cause additional consumption of reservoir storage, and leads to great reliance on hydrosuction to reservoir desilt, degradation of downstream environment and inefficient utilization of water resources. This paper elaborately creates an opportune chance for potential empty flushing under such high-pressure of water supply. If this pressure can be somehow relieved, the practical benefits of the proposed method could be more evident, since all the problems stem

from the same core: insufficiency of available water with acceptable quality for all purposes. While the operators are forced to myopically prevent the imminent water shortage risks, reservoir sedimentation also imposes equivalent and long-term threat to the degeneration of water supply yield. The urgent needs of both desilting and water supply may also endow a new role to the conventional projects of water resources development. In addition to elevating

the yield of water supply, it may exploit more water to allow recovery and enhanced desilting





of existing reservoirs, thus allowing the entire system to advance toward the goal of sustainability.

**Data availability**

5     The data used in this study, including 46 years of daily hydrological series, simulated water allocation process, as well as hundreds of input and output files of the GWASIM and SRH-2D models, are available from the authors upon requests (chiawenwu1977@gmail.com). The executive file of SRH-2D model and the associated manuals are available on the website of U.S. Bureau of Reclamation.

10     (https://www.usbr.gov/tsc/techreferences/computer%20software/models/srh2d/index.html).

**Author contribution**

    The study was initiated by FNFC, who also got the funds. FNFC and CWW both contribute to the methodological development of simulation and optimization models. CWW 15 carried out the contents of most analysis in the case study. The numerical simulation of SRH 2D was performed by FZL. The manuscript was jointly drafted by CWW and FNFC.

**Competing interests**

    The authors declare that they have no conflict of interest.

**Acknowledgement**

    This work was jointly supported by the Ministry of Science and Technology (Grant No. NSC102-2221-E006-179) and Water Resources Agency (No. MOEA/WRA/101-WRASB-03), Taiwan, R.O.C. The authors acknowledge the Technical Service Center, US Bureau of 25 Reclamation for the numerical software provided for this study. We also appreciate Dr. Jihn-



Sung Lai, the research fellow of Hydrotech Research Institute, National Taiwan University for providing the sediment transport knowledge and suggestions on simulation implementation.

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
