# Peer review of "Minimizing the impact of vacating instream storage of a multi-reservoir system: a tradeoff study of water supply and empty flushing"

_Hydrology and Earth System Sciences, 2020_

## Referee Comment (RC1) · Anonymous Referee #1 · 21 Sep 2020

GENERAL COMMENTS

This paper explores the feasibility of empty flushing in a two-reservoirs system, minimizing the impact of the operation on the multiple use of water storage (i.e., municipal, agricultural, industrial, and hydropower supply). The reservoirs are located in SW Taiwan. Due to siltation, the larger one (Tsengwen Reservoir) has lost about 30% of its original storage capacity (630 Mm3) in the 45 years following dam closure (1973). The smaller one (Wushanto Reservoir), if properly managed, could satisfy local water demand when flushing Tsengwen Reservoir. The general subject of adopting empty flushing to recover reservoir storage, and the specific topic of optimizing multi-purpose

systems comprising several reservoirs deserve in my opinion the interest of the international scientific literature dealing with water resources management. I think that the manuscript could be significantly improved by considering comments and suggestions provided below.

SPECIFIC COMMENTS

1. In my opinion, the paper is overly long. To improve its readability, I suggest to shortening or moving to Appendixes or Supplementary Material sections and paragraphs of minor importance relatively to the main objective of the study. Examples are reported below.

- Table 1 might be moved to Appendix/Supplementary Material.

- The estimation of parameter psi of Equation 1 (P35-P38), including data from further reservoirs, might be moved to Appendix/Supplementary Material.

- The sensitivity analysis (P45-46) might be moved to Appendix/Supplementary Material. By the way, the linear variation of the desilting volume with psi could have been expected, due to the structure of Equation1.

- The 2D simulations of sediment transport throughout the drawn-down reservoir during empty flushing (P 47-48) can be moved to Appendix/Supplementary Material. From the one hand, I think that the efforts made by the Authors to validating the adopted psi value are commendable. However, the proper presentation of these simulations would require additional space, overloading the paper.

2. Section Results and Discussion contains several elements related to the description of the investigated system and of the adopted methodology. I suggest moving these paragraphs to the Methods section. Examples are reported below.

- Water demand of the system and inflow to Tsengwen Reservoir (P29, Figure 5) are not results, and can go to the subsection describing the case study.

- The scheme of the system (Figure 6) is not a result, and can go to the subsection describing the case study.

- The modified balancing curves (P31, Figure 7) are not results, and would be moved to the Methods section.

- The methodology to assessing the impact of empty flushing on the short-term water supply (P38-39) is not a result, and would be moved to the Methods section.

- The refinement through Equations 12 and 13 (P41) is not a result, and would be moved to the Methods section.

3. Partly connected to previous point 2: though the adopted methodology could be extended to multi-reservoirs systems, it was developed (I would say "tailored") on the two-reservoirs study case. I therefore suggest to fully describing the investigated system in a dedicated subsection, and to describing the adopted methodology with specific reference to the case study. Later on, in the Discussion section, the Authors can comment on the possible extension to different multi-reservoirs systems. In the current version of the manuscript, information concerning the investigated system is fragmented over different paragraphs, thus confusing the reader.

4. The calibration of the model throughout the period 1974-2009 and its validation in the following years (2010-2017 – Par. 3.7) is rather unclear to me: I expected some comparison of simulated vs observed data, but I did not noticed it. Perhaps this point might be clarified.

5. The proposed strategy poses some risks (shortage at least, but also hydraulic and environmental issues to the river section below Tsengwen Reservoir). In contrast, the predicted contribution to desilting Tsengwen Reservoir is low (Figure 18). Accordingly, current management adopts hydro-suction, downstream settling, and removal by flood spilling (P24-25). Moreover, the capacity to inflow ratio for Tsengwen would seem very high (in the range 4-5, depending on the adopted storage, original or current),

but my estimate of CIR can be affected by wrong inflow data provided in Par. 2.4 (see Technical Corrections below). I think that these arguments would be properly addressed and deeply discussed in the revised version of the paper.

6. The environmental impact of empty flushing has, in my opinion, marginal relevance in the proposed strategy. Rather than presenting it as automatically addressed (Par. 2.3, and particularly P23, L18-20), I would comment it the Discussion section (possibly including more recent references) as a potential source of further constraints. In fact, when considering the impact of sediment flushing on downstream biotas, limits on suspended sediment concentration and dissolved oxygen (as well as on streambed aggradation) should be accounted for.

TECHNICAL CORRECTIONS

P3, L5: "extraordinary water quality" is unclear, please rephrase.

P12, L20: "capacitated" is unclear, please rephrase.

Figures writing is frequently too small and could be enlarged to improve readability (see Figures 1, 5, 6, and 13).

P19, L6: in order to avoid confusion with sediment, water can be specified before "volume".

P19, L9: did you mean refill instead of "fulfill"?

P26, L13: I would remove "experimental setup".

P26, L20: I think that 120 Mm3 annual inflow is too small, suggest checking this (very important) parameter (see previous point 5).

P28, L13: replace "result" by results.

P36, L23-24: how did you get volumetric concentrations? The adopted sediment density is not specified.

---

## Referee Comment (RC2) · Anonymous Referee #2 · 2 Nov 2020

General comments

The paper "Minimizing the impact of vacating instream storage of a multi-reservoir system: a tradeoff study of water supply and empty flushing" describes a modelling framework simulate sediment flushing in reservoir and to derive the optimal dam water release strategy to guarantee adequate sediment flushing without excessively hindering water availability, in river systems where along with the primary reservoir there are others that can be used to reduce the water scarcity when the flushing is in progress and during the subsequent refilling. Both the timing and the volumes of water release are taking into consideration, as well as how the different operating strategies of the

dams in the network interact among each other to avoid water scarcity while allowing for sediment removal in reservoirs. In general, I found the paper to be well written and scientifically sound; the new methodology is described in detail, and the reasoning behind each assumption and parameter is clearly stated and explained. The model is then applied on the case study of the Tsengwen and Wushanto Reservoirs, in southern Taiwan. The different combinations of optimal release and flushing strategies are adequately explored, and the results shown are solid, and reinforced by a sensitivity analysis of the results for a parameter on the transport capacity equation, a numerical simulation of the flushing process to validate the effectiveness of the flushing and a validation of an optimal flushing strategy selected on a time period different to the one used for calibration. However, I believe the paper suffers from a general lack of focus in the first part, where the methodology is presented, and some shortcomings in the application of the framework on the case study.

Specific comments

To start, I believe the case study application should be cited both in the abstract and in the introduction. In section 2 multiple case study are cited, including the Tsengwen and Wushanto Reservoirs, that is however not reported as the main case study. The objective of the author might be to present the methodology in broader way possible, in order to highlight its flexibility and general nature, but I still believe that it should be made clear to the reader which of the numerous cited reservoirs are used for the application of the framework, from the beginning of the paper.

I believe section 2.1 needs a general rewrite, as I think the number of parameters, case studies, example and lead to a lack of focus and damage the readability of the paper. For example, I believe table 1 to be superfluous in this case, as the numerous examples of flushing in the tables are not properly commented and do not benefit the overall narrative of the section, and so they should be moved to the supplementary material or remover altogether and substituted with proper references. Likewise, I would also remove table 3 and 4 in section 3 and just leave the relative references (table 4 is not

even referenced in the paper).

I think section 2.3 should be greatly reduced or altogether removed and integrated into the conclusion section. While the environmental effects of empty flushing are definitely worth considering, they are not the focus of the paper and are not integrated in the analysis of the optimal strategy in the case study application. Given its length, section 2.3 may give the impression to the reader that the downstream environment protection is one of the objectives formalized in the search for the optimal flushing strategy, which it is not. The impact on the downstream environment is only brought back in a small section on page 50, not enough to justify the presence of section 2.3.

Regarding the application on the case study, I think one aspect that should be considered would be the simulation of the hydrological conditions not considered in the studied timeframe (1975-2017). In particular, I think the approach would benefit from the analysis of the objective performances under synthetically generated annual hydrological series with extreme events, both floods and drought, confronting the performances with or without flushing. In particular, droughts are of particular concern in this case, as shown in figure 17. I believe this point should be explored further, as it is it seems the obtained solutions do not perform well under for the water supply during period with unexpected lack of floods.

I think the results shown in fig. 18 should be commented further. From the figure, it looks like employing the optimal empty flushing strategy in the past would have led to a desilted volume of approximately half a million m3. I think it should be given a framework to the reader to evaluate if this value is low or high, comparing it to the increased water scarcity. Moreover, I would also show in this figure the trajectories for the other optimal strategies reported in table 5, as I believe it would be far more explicative than the data reported in the table.

---

## Author Comment (AC1) · 3 Dec 2020

General response: We sincerely appreciate the efforts of the reviewer to scrutinize the manuscript. The authors generally agree with the reviewer and will dedicate to revise the manuscript accordingly. The major revisions expected to address the reviewers' comments are summarized as the following: 1.The structure of the article will be re-organized as the following sections: (1)introduction, (2) qualitative conditions to implement empty flushing, (3) the case study area, (4)the adopted methods, (5)results and discussion, (6)potential future extension and (7)conclusion remarks. 2.The description in the introduction and methodology sections will focus more on the specific

schematic of the case study system, and the extension to other general schematics will be moved to and more precisely addressed in the 6-th section. 3.All the materials about the supplemented data and references, including the field and numerical validation of the estimation of volume of flushed sediments, will be removed from the manuscript and provided in additional supplemented materiel files. 4.The description regarding the impact on downstream environment will be limited and shortened in the 5-th and 6-th sections with updated references. 5.The potential risks imposed by emptying reservoir on the following water supply and measures to alleviate or even offset the incremental water shortage will be more thoroughly presented in the 5-th section. 6.All the technical corrections mentioned by reviewers will be modified, improved and clarified in the revised version of the manuscript.

Point-by-point responses to the specific comments: 1.To recentralize the presentation on the theme of the research, all the suggestions by the reviewer will be undertaken accordingly.

2.The distributed information of the case study system will be gathered and integrated in the 3-rd section. The link between the case study area and the adopted method will also be more clearly and specifically explained.

3.The sections of the article will be reorganized as described in the general response.

4.The simulation model is designed to evaluate the performance of a water resources system under specific storage volume, water demand and operating rules. The simulation requires sequential daily routing of system operation for several decades of inflow series to reflect the long-term hydrological variation. Based on this aspect, comparing the simulation results with historical operating records may induce misinterpretations, since the reservoir storage and water demands were not stationary during the historical periods. The calibration analysis in the paper does not tune parameters related to physical movement process of water or sediment. Instead, it calibrates the optimal operating rules for the simulating duration. The validation is then testing the rules using

the model with inflow series outside of the calibration timeframe to check its validity for general conditions. These points will be added in the associated sections in the revised manuscript.

5.The annual inflow volume to the Twengwen Reservoir in the original manuscript is a typo and should be corrected as 1.2 billion m3. This leads to a relatively small CIR ratio for the Twengwen Reservoir, and the general principles in literatures will not recommend the reservoir to implement empty flushing. Nonetheless, over 50% of the inflow volume concentrates within some significant flood events for Twengwen Reservoir. In addition, the presence of downstream off-line smaller reservoir adequately ensures short-term stable water supply, if properly managed. These conditions inspire the authors to elaborately create the opportune chances for potential empty flushing of Twengwen Reservoir, which suffers severely from both water insufficiency and siltation.

Except the shortage risk, any hydraulic-based desilting means impose impacts on the downstream river sections, including the currently adopted hydro-suction operation. Adequate flood spillage is a necessary condition for the effective removal of downstream deposited sediments. This condition might not be met during years without significant flood events, following which the hydro-suction operation will be halted and the impact on the depositing section of the river will last until the next adequate reservoir spillage. Nonetheless, the urgent need of achieving balance between annual inflowing and removing sediments require all the desilting means to cooperate rather than competing with each other. There are no conflicts between empty flushing, hydro-suction and sediment sluicing, as long as the shortage risk imposed by the first can be properly contained.

As for the incremental shortage risk, the problem comes from the rare situation while the frontal-induced inflow in the early flood season is abundant and the following invading typhoons are entirely absence for the remaining 5 months. Thus the water released for empty flushing cannot be recovered and incremental shortage is created. Nonetheless, this rare condition would inevitably lead to large scale suspension of the

first semiannual irrigation, whether empty flushing in the previous year is performed or not. With or without empty flushing, the water originally supplied to the first semiannual irrigation, the volume of which ranges between 0.2∼0.3 billion m3, will be kept to secure public water supply. The annual demand of public purpose of this system is only 0.12 billion m3 and the empty flushing consumes water under 0.09 billion m3 according to the simulation. This shows the risk of increased shortage induced by empty flushing for this particular situation will be completely offset in reality.

In the last paragraph of the conclusion section, we do address that: "The high risk of water shortage in the case study area currently dictates the operating objective to solely focus on reliable water supply. This restricts the feasibility of not only empty flushing, but any other operations may cause additional consumption of reservoir storage, and leads to great reliance on hydrosuction to reservoir desilt, degradation of downstream environment and inefficient utilization of water resources. If this pressure can be somehow relieved, the practical benefits of the proposed method could be more evident, since all the problems stem from the same core: insufficiency of available water with acceptable quality for all purposes. While the operators are forced to myopically prevent the imminent water shortage risks, reservoir sedimentation also imposes equivalent and long-term threat to the degeneration of water supply yield. The urgent needs of both desilting and water supply may also endow a new role to the conventional projects of water resources development. In addition to elevating the yield of water supply, it may exploit more water to allow recovery and enhanced desilting of existing reservoirs, thus allowing the entire system to advance toward the goal of sustainability."

In addition, the first sentence in the same paragraph is considered by the authors as the major step forward from the current disciplines of both reservoir desilting and water resources management: "Integrating reservoir desilting considerations with water supply operation creates more facets into the multi-objective water resources management. In addition to irrigation, municipal, industrial and hydropower purposes, the competition

of water extends to include sediment flushing, sluicing, vacating previous dredged and deposited sediments, and alleviating their impacts on downstream environment."

6.The authors agree with the reviewer and the content about the environmental impact will be more properly presented in the suggested section with shortened length in the revised manuscript.

———————————————————

---

## Author Comment (AC2) · 3 Dec 2020

The general response is as in the reply to the anonymous referee #1.

Point-by-point responses to the specific comments: 1.According to the comments by both reviewers, the structure of the article will be reorganized. The introduction section will be reformed to specifically present the case study system. The case study section is moved in front of the methodology to enhance their linking and avoid divergence. As for the comments about "which of the numerous cited reservoirs are used for the application of the framework", other than the presented case study, we did not see similar implementation in the literature to explicitly address the trade-off of empty reservoir

storage and maintaining water supply in a multi-reservoir system.

2.All the materials about the supplemented data and references, including the field and numerical validation of the estimation of volume of flushed sediments, will be removed from the manuscript and provided in additional supplemented materiel files.

3.In order to recentralize the presentation on the theme of the research, all the suggestions will be undertaken accordingly.

4.The main idea of this study is to jointly operate multiple reservoirs to create opportunities for empty flushing without excessively hindering water supply. The structure of a multi-reservoir system essentially comprises of multiple inflow series to the system. For example, three daily inflow records at separate control points, each with duration more than 4 decades, are in presence in the case study system. This imposes difficulties in synthetically generating hydrological series since the multi-correlation among daily inflows of different sites should be properly modeled in order to correctly represent the temporal and spatial stochastic hydrological nature. According to the recent operating experiences, the return period of extreme drought leading to large scale suspension of irrigation in southern Taiwan is approximately 10 years. The current simulation time span of 43 years should already be adequate to accommodate this frequency of drought occurrence. Further, the insufficiency of inflow during extreme droughts in the past is actually due to the absences of typhoon-induced floods in the previous wet season. This requires simulating the scale, frequency and duration of flood events induced by different weather factors, along with continuous hydrological modeling of the following long-term recession process. All these factors increase the challenges in synthetically generating hydrological series and overload the paper beyond the scope of its theme.

The obtained solutions of initiating and terminating an empty flushing operation inevitably impose risks for water supply, though the research strives to control the incremental risk within manageable range while maximizing performances of sediment

flushing. Nonetheless, it is very difficult, if not impossible, to control the water shortage in the next entire year merely by determining whether to implement or suspend an empty flushing operation of several days. In the case study area, 90% of the average annual inflow occurs during May to October, and only 15% of which occurs during the prescribed feasible periods for empty flushing. So that when encountering abundant inflow in the early flood season, no one can foresee that the expected following typhoon-induced floods will be entirely absencet for the remaining 5 months. This rare situation did occur twice in the flood seasons of 2014 and 2020. Thus the water released for empty flushing in the simulation cannot be recovered and incremental shortage created. Nonetheless, this condition always leads to large scale suspension of the first semiannual irrigation, whether empty flushing in the previous year is performed or not. The water originally supplied to the first semiannual irrigation, the volume of which ranges between 0.2~0.3 billion m3, will be kept to ensure security of public water supply. The annual demand of public purpose is only 0.12 billion m3 and the empty flushing consumes water under 0.09 billion m3 according to the simulation. This shows the risk of increased shortage induced by empty flushing for this particular situation will be completely offset in reality. For the other regular empty flushing events, the incremental shortage only concentrates before the following floods, after which the released water can be recovered, and can be managed by postponing the second semiannual irrigation if necessary. The above discussion will be properly integrated into the revised manuscript.

5. (1)According to the data from the government, the estimated average annual inflowing sediment volume for Twengwen Reservoir is 5.6 million m3 per year. a. With the newly-constructed desilting tunnel with capacity as 1,000 m3/s, the annual volume by sediment venting during floods is estimated as 1.60 million m3 per year. b. The desilting volume of hydro-suction is currently increased to 3.15 million m3 per year. c. While several watershed management measures are expected to reduce the inflowing sediment volume by 0.5 million m3 per year, there is still a gap of 0.35 million m3 per year to be desilt by mechanical excavation. d. The cost for mechanical excavation is

approximately 15~18 USD per desilt volume (m3) and can be replaced by the empty flushing with manageable incremental water shortage. e. More aggressive flushing operations accommodate the uncertainties of effectiveness of watershed management measures, spillage-required sluicing and removal of hydrosuction deposition, and potentially recover the siltation volume of reservoir.

(2)Figure 18 and Table 5 will be integrated in the revised manuscript according to the precise point by the reviewer.

---

## Author Response (AR1)

General Responses by the Authors

We sincerely appreciate the efforts of reviewers to scrutinize the manuscript. Major contributions, including the relevancy of the subject, the well-written and scientifically-sound contents, the elaboration of the adopted methods with adequately-explored case study application, were identified along with some drawbacks regarding the presentation of the paper. The authors generally agree with the current reviewers and have dedicated to revise the manuscript accordingly. The major revisions to address the reviewers' comments are summarized as the following:

1. The structure of the article is re-organized as the following sections: 1. introduction, 2. the case study area, 3. the qualitative conditions to implement empty flushing and the adopted methods, 4. results and discussion, 5. potential future extension and 6. conclusion remarks.

2. The description in the introduction and methodology sections focuses more on the specific schematic of the case study system, and the extension to other general schematics is moved to and more precisely addressed in the 5th section.

3. All the materials about the supplemented data and references, including the field and numerical validation of the estimation of volume of flushed sediments, are removed from the manuscript and provided in appendices.

4. The description regarding the impact on downstream environment is shortened and moved to the 5th section with updated references.

5. The potential risks imposed by emptying reservoir on the following water supply and measures to alleviate or even offset the incremental water shortage are more thoroughly presented in the sub-sections 4.4 and 4.5.

6. All the technical corrections mentioned by reviewers are modified, improved and clarified in the revised version of the manuscript.

Detailed point-by-point responses are listed as below

| Reviewer # 1 | |
|---|---|
| General Comments | Authors' response |
| This paper explores the feasibility of empty flushing in a two-reservoirs system, minimizing the impact of the operation on the multiple use of water storage (i.e., municipal, agricultural, industrial, and hydropower supply). The reservoirs are located in SW Taiwan. Due to siltation, the larger one (Tsengwen Reservoir) has lost about 30% of its original storage capacity (630 Mm$^3$) in the 45 years following dam closure (1973). The smaller one (Wushanto Reservoir), if properly managed, could satisfy local water demand when flushing Tsengwen Reservoir. The general subject of adopting empty flushing to recover reservoir storage, and the specific topic of optimizing multi-purpose systems comprising several reservoirs deserve in my opinion the interest of the international scientific literature dealing with water resources management. I think that the manuscript could be significantly improved by considering comments and suggestions provided below. | The reviewer's recognition of the relevancy of the subject is very much appreciated. The valuable comments and suggestions are all integrated into the revised manuscript. |
| Specific comments | Authors' response |
| 1. In my opinion, the paper is overly long. To improve its readability, I suggest to shortening or moving to Appendixes or Supplementary Material sections and paragraphs of minor importance relatively to the main objective of the study. Examples are reported below.
- Table 1 might be moved to Appendix/Supplementary Material.

- The estimation of parameter psi of Equation 1 (P35-P38), including data from further reservoirs, might be moved to Appendix/Supplementary Material.

- The sensitivity analysis (P45-46) might be moved to Appendix/Supplementary Material. By the way, the linear variation of the desilting volume with psi could have been expected, due to the structure of Equation1.

- The 2D simulations of sediment transport throughout the drawn-down reservoir during empty flushing (P 47-48) can be moved to Appendix/Supplementary Material. | The growing length of the article is a result of attempting to address and integrate all comments from previous reviews.

In order to recentralize the presentation on the theme of the research, all the suggestions by the reviewer are undertaken accordingly.

- The original Table 1 is moved to Appendix 1
- The estimation of parameter psi of Equation 1 (P35-P38), including data from further reservoirs, are moved to Appendix 2.
- The sensitivity analysis is moved to Appendix 4.
- The numerical simulation is moved to Appendix 3. |

| | |
|---|---|
| From the one hand, I think that the efforts made by the Authors to validating the adopted psi value are commendable. However, the proper presentation of these simulations would require additional space, overloading the paper. | |
| 2. Section Results and Discussion contains several elements related to the description of the investigated system and of the adopted methodology. I suggest moving these paragraphs to the Methods section. Examples are reported below.

- Water demand of the system and inflow to Tsengwen Reservoir (P29, Figure 5) are not results, and can go to the subsection describing the case study.

- The scheme of the system (Figure 6) is not a result, and can go to the subsection describing the case study.

- The modified balancing curves (P31, Figure 7) are not results, and would be moved to the Methods section.

- The methodology to assessing the impact of empty flushing on the short-term water supply (P38-39) is not a result, and would be moved to the Methods section.

- The refinement through Equations 12 and 13 (P41) is not a result, and would be moved to the Methods section. | The associated revisions are made accordingly.

- -Water demand of the system and inflow to Tsengwen Reservoir are moved to the 2nd section describing the case study system.
- The scheme of the system is moved to the 2nd section.
- The modified balancing curves are moved to the subsection 3.2.1.
- The methodology to assessing the impact of empty flushing on the short-term water supply are integrated into the original content in the subsection 3.2.5. |
| 3. Partly connected to previous point 2: though the adopted methodology could be extended to multi-reservoirs systems, it was developed (I would say "tailored") on the two reservoirs study case. I therefore suggest to fully describing the investigated system in a dedicated subsection, and to describing the adopted methodology with specific reference to the case study. Later on, in the Discussion section, the Authors can comment on the possible extension to different multi-reservoirs systems. In the current version of the manuscript, information concerning the investigated system is fragmented over different paragraphs, thus confusing the reader. | The sections of the article are reorganized as following: 1. introduction, 2. the case study area, 3. qualitative conditions to implement empty flushing, and the adopted methods, 4. results and discussion, 5. potential future extension and 6. conclusion remarks.

The distributed information of the case study system is gathered and integrated in the 2nd section.
The link between the case study area and the adopted method is more clearly and specifically explained in |

| | the 1st and 3rd sections. |
|---|---|
| 4. The calibration of the model throughout the period 1974-2009 and its validation in the following years (2010-2017 – Par. 3.7) is rather unclear to me: I expected some comparison of simulated vs observed data, but I did not notice it. Perhaps this point might be clarified. | The simulation model is designed to evaluate the performance of a water resources system under specific storage volume, water demand and operating rules. The simulation requires sequential daily routing of system operation for several decades of inflow series to reflect the long-term hydrological variation. Based on this aspect, comparing the simulation results with historical operating records may induce misinterpretations, since the reservoir storage and water demands were not stationary during the historical periods.

The calibration analysis in the paper does not tune parameters related to physical movement process of water or sediment. Instead, it calibrates the optimal operating rules for the simulating duration. The validation is then testing the rules using the model with inflow series outside of the calibration timeframe to check its validity for general conditions. These points are added in subsections 3.2.4 and 4.5 in the revised manuscript. |
| 5. The proposed strategy poses some risks (shortage at least, but also hydraulic and environmental issues to the river section below Tsengwen Reservoir). In contrast, the predicted contribution to desilting Tsengwen Reservoir is low (Figure 18). Accordingly, current management adopts hydro-suction, downstream settling, and removal by flood spilling (P24-25). Moreover, the capacity to inflow ratio for Tsengwen would seem very high (in the range 4-5, depending on the adopted storage, original or current), but my estimate of CIR can be affected by wrong inflow data provided in Par. 2.4 (see Technical Corrections below). I think that these arguments would be properly addressed and deeply discussed in the revised version of the paper. | The annual inflow volume to the Twengwen Reservoir in the original manuscript is a typo and is corrected as 1.2 billion m3. This leads to a CIR ratio as 0.38 for the Twengwen Reservoir, and the general principles in literatures will not recommend the reservoir to implement empty flushing. Nonetheless, over 50% of the inflow volume concentrates within some significant flood events for Twengwen Reservoir. In addition, the presence of downstream off-line smaller reservoir adequately ensures short- |

term stable water supply, if properly managed. These conditions inspire the authors to elaborately create the opportune chances for potential empty flushing of Twengwen Reservoir, which suffers severely from both water insufficiency and siltation.

Except the shortage risk, any hydraulic-based desilting means impose impacts on the downstream river sections, including the currently adopted hydrosuction operation. Adequate flood spillage is a necessary condition for the effective removal of downstream deposited sediments. This condition might not be met during years without significant flood events, following which the hydrosuction operation will be halted and the impact on the depositing section of the river will last until the next adequate reservoir spillage. Nonetheless, the urgent need of achieving balance between annual inflowing and removing sediments require all the desilting means to cooperate rather than competing with each other. There are no conflicts between empty flushing, hydrosuction and sediment sluicing, as long as the shortage risk imposed by the first can be properly contained.

As for the incremental shortage risk, the problem comes from the rare situation while the frontal-induced inflow in the early flood season is abundant and the following invading typhoons are all insignificant. Thus the water released for empty flushing cannot be recovered and incremental shortage is created. Nonetheless, this rare condition

would inevitably lead to large scale suspension of the first semiannual irrigation, whether empty flushing in the previous year is performed or not. With or without empty flushing, the water originally supplied to the first semiannual irrigation, the volume of which ranges between 0.2~0.3 billion $m^3$, will be kept to ensure security of public water supply. The annual demand of public purpose of this system is only 0.12 billion $m^3$ and the empty flushing consumes water under 0.09 billion $m^3$ according to the simulation. This shows the risk of increased shortage induced by empty flushing for this particular situation will be completely offset in reality.

In the last paragraph of the conclusion section, we do address that:
"The high risk of water shortage in the case study area currently dictates the operating objective to solely focus on reliable water supply. This restricts the feasibility of not only empty flushing, but any other operations may cause additional consumption of reservoir storage, and leads to great reliance on hydrosuction to reservoir desilt, degradation of downstream environment and inefficient utilization of water resources. If this pressure can be somehow relieved, the practical benefits of the proposed method could be more evident, since all the problems stem from the same core: insufficiency of available water with acceptable quality for all purposes. While the operators are forced to myopically prevent the imminent water shortage risks, reservoir sedimentation also imposes equivalent and long-term

| | threat to the degeneration of water supply yield. The urgent needs of both desilting and water supply may also endow a new role to the conventional projects of water resources development. In addition to elevating the yield of water supply, it may exploit more water to allow recovery and enhanced desilting of existing reservoirs, thus allowing the entire system to advance toward the goal of sustainability."

In addition, the first sentence in the same paragraph is considered by the authors as the major step forward from the current disciplines of both reservoir desilting and water resources management: "Integrating reservoir desilting considerations with water supply operation creates more facets into the multi-objective water resources management. In addition to irrigation, municipal, industrial and hydropower purposes, the competition of water extends to include sediment flushing, sluicing, vacating previous dredged and deposited sediments, and alleviating their impacts on downstream environment." |
|---|---|
| 6. 6. The environmental impact of empty flushing has, in my opinion, marginal relevance in the proposed strategy. Rather than presenting it as automatically addressed (Par. 2.3, and particularly P23, L18-20), I would comment it the Discussion section (possibly including more recent references) as a potential source of further constraints. In fact, when considering the impact of sediment flushing on downstream biotas, limits on suspended sediment concentration and dissolved oxygen (as well as on streambed aggradation) should be accounted for. | The content is more properly presented in the 5th section with shortened length in the revised manuscript. |
| TECHNICAL CORRECTIONS | Authors' response |
| 1.P3, L5: "extraordinary water quality" is unclear, please rephrase. P12, L20: "capacitated" is unclear, please rephrase. | 1.The phrases "extraordinary water quality" and "capacitated bottom outlets" mentioned by the reviewers are all deleted. |

| | |
|---|---|
| 2.Figures writing is frequently too small and could be enlarged to improve readability (see Figures 1, 5, 6, and 13). | 2.The original Figs. 1,5,6 and 13 are redrawn and improved as the Figs. 6, 5, 2 and 15 in the revised manuscript. |
| 3.P19, L6: in order to avoid confusion with sediment, water can be specified before "volume". | 3&4. The suggested words have been added, as in the subsection 3.2.4. |
| 4.P19, L9: did you mean refill instead of "fulfill"? | |
| 5.P26, L13: I would remove "experimental setup". | 5. The title has been modified accordingly, as the title of section 2 in the revised manuscript. |
| 6.P26, L20: I think that 120 Mm3 annual inflow is too small, suggest checking this (very important) parameter (see previous point 5). | 6. The original data is a typo and has been modified into 1.2 billion $m^3$, as described in subsection 2.1. |
| 7.P28, L13: replace "result" by results. | 7. The title has been modified as "Analysis, results and discussion" of section 4 in the revised manuscript. |
| 8.P36, L23-24: how did you get volumetric concentrations? The adopted sediment density is not specified. | 8. The desilting volume is converted from the estimated flushing discharge with bulk density as 1.56 $T/m^3$. This sentence is added in subsection 4.3. |

Reviewer # 2

| General Comments | Authors' response |
|---|---|
| The paper "Minimizing the impact of vacating instream storage of a multi-reservoir system: a tradeoff study of water supply and empty flushing" describes a modelling framework simulate sediment flushing in reservoir and to derive the optimal dam water release strategy to guarantee adequate sediment flushing without excessively hindering water availability, in river systems where along with the primary reservoir there are others that can be used to reduce the water scarcity when the flushing is in progress and during the subsequent refilling. Both the timing and the volumes of water release are taking into consideration, as well as how the different operating strategies of the dams in the network interact among each other to avoid water scarcity while allowing for sediment removal in reservoirs. In general, I found the paper to be well written and scientifically sound; the new methodology is described in detail, and the reasoning behind each assumption and parameter is clearly stated and explained. The model is then applied on the case study of the Tsengwen and Wushanto Reservoirs, in southern Taiwan. The different combinations of optimal release and flushing strategies are adequately explored, and the results shown are solid, and reinforced by a sensitivity analysis of the results for a parameter on the transport capacity equation, a numerical simulation of the flushing process to validate the effectiveness of the flushing and a validation of an optimal flushing strategy selected on a time period different to the one used for calibration. However, I believe the paper suffers from a general lack of focus in the first part, where the methodology is presented, and some shortcomings in the application of the framework on the case study. | The approval of the contents, proposed methods, case study results, validation analysis...etc., by the reviewer is very much appreciated. The mentioned drawbacks and shortcomings are fully addressed in the revised manuscript. |
| Specific comments | Authors' response |
| To start, I believe the case study application should be cited both in the abstract and in the introduction. In section 2 multiple case study are cited, including the Tsengwen and Wushanto Reservoirs, that is however not reported as the main case study. The objective of the author might be to present the methodology in broader way possible, in order to highlight its flexibility and general nature, but I still believe that it should be made clear to the reader which of the numerous cited reservoirs are used for the application of the framework, from the beginning of the paper | According to the comments by both reviewers, the structure of the article is reorganized. The introduction section is reformed to specifically present the case study system. The case study section is moved in front of the methodology to enhance their linking and avoid divergence. |

| | As for the comments about "which of the numerous cited reservoirs are used for the application of the framework", other than the presented case study, we did not see similar implementation in the literature to explicitly address the trade-off of empty reservoir storage and maintaining water supply in a multi-reservoir system. |
|---|---|
| I believe section 2.1 needs a general rewrite, as I think the number of parameters, case studies, example and lead to a lack of focus and damage the readability of the paper. For example, I believe table 1 to be superfluous in this case, as the numerous examples of flushing in the tables are not properly commented and do not benefit the overall narrative of the section, and so they should be moved to the supplementary material or remover altogether and substituted with proper references. Likewise, I would also remove table 3 and 4 in section 3 and just leave the relative references (table 4 is not even referenced in the paper). | All the materials about the supplemented data and references, including the field and numerical validation of the estimation of volume of flushed sediments, are moved to the Appendices. |
| I think section 2.3 should be greatly reduced or altogether removed and integrated into the conclusion section. While the environmental effects of empty flushing are definitely worth considering, they are not the focus of the paper and are not integrated in the analysis of the optimal strategy in the case study application. Given its length, section 2.3 may give the impression to the reader that the downstream environment protection is one of the objectives formalized in the search for the optimal flushing strategy, which it is not. The impact on the downstream environment is only brought back in a small section on page 50, not enough to justify the presence of section 2.3. | In order to recentralize the presentation on the theme of the research, all the suggestions are undertaken accordingly, as the 5th section in the revised manuscript. |
| Regarding the application on the case study, I think one aspect that should be considered would be the simulation of the hydrological conditions not considered in the studied timeframe (1975-2017). In particular, I think the approach would benefit from the analysis of the objective performances under synthetically generated annual hydrological series with extreme events, both floods and drought, confronting the performances with or without flushing. In particular, droughts are of particular concern in this case, as shown in figure 17. I believe this point should be explored further, as it is it seems the obtained solutions do not perform well under for the water supply during | The main idea of this study is to jointly operate multiple reservoirs to create opportunities for empty flushing without excessively hindering water supply. The structure of a multi-reservoir system essentially comprises of multiple inflow series to the system. For example, three daily inflow records at separate control points, each with duration more than 4 decades, are in presence in |

period with unexpected lack of floods.

the case study system. This imposes difficulties in synthetically generating hydrological series since the multi-correlation among daily inflows of different sites should be properly modeled in order to correctly represent the temporal and spatial stochastic hydrological nature.

According to the recent operating experiences, the return period of extreme drought leading to large scale suspension of irrigation in southern Taiwan is approximately 10 years. The current simulation time span of 43 years should already be adequate to accommodate this frequency of drought occurrence. Further, the insufficiency of inflow during extreme droughts in the past is actually due to the absences of typhoon-induced floods in the previous wet season. This requires simulating the scale, frequency and duration of flood events induced by different weather factors, along with continuous hydrological modeling of the following long-term recession process. All these factors increase the challenges in synthetically generating hydrological series and overload the paper beyond the scope of its theme.

The obtained solutions of initiating and terminating an empty flushing operation inevitably impose risks for water supply, though the research strives to control the incremental risk within manageable range while maximizing performances of sediment flushing. Nonetheless, it is very difficult, if not impossible, to control the water shortage in the next entire year merely by determining whether to

implement or suspend an empty flushing operation of several days. In the case study area, 90% of the average annual inflow occurs during May to October, and only 15% of which occurs during the prescribed feasible periods for empty flushing. So that when encountering abundant inflow in the early flood season, no one can foresee that the expected following typhoon-induced floods will be insignificant for the remaining 5 months. This rare situation did occur twice in the flood seasons of 2014 and 2020. Thus the water released for empty flushing in the simulation cannot be recovered and incremental shortage created. Nonetheless, this condition always leads to large scale suspension of the first semiannual irrigation, whether empty flushing in the previous year is performed or not. The water originally supplied to the first semiannual irrigation, the volume of which ranges between 0.2~0.3 billion $m^3$, will be kept to ensure security of public water supply. The annual demand of public purpose is only 0.12 billion $m^3$ and the empty flushing consumes water under 0.09 billion $m^3$ according to the simulation. This shows the risk of increased shortage induced by empty flushing for this particular situation will be completely offset in reality. For the other regular empty flushing events, the incremental shortage only concentrates before the following floods, after which the released water can be recovered, and can be managed by postponing the second semiannual irrigation if necessary.

| | The above discussion are integrated into subsection 4.5 and section 5 in the revised manuscript. |
|---|---|
| I think the results shown in fig. 18 should be commented further. From the figure, it looks like employing the optimal empty flushing strategy in the past would have led to a desilted volume of approximately half a million m$^3$. I think it should be given a framework to the reader to evaluate if this value is low or high, comparing it to the increased water scarcity. Moreover, I would also show in this figure the trajectories for the other optimal strategies reported in table 5, as I believe it would be far more explicative than the data reported in the table. | 1. According to the data from the government, the estimated average annual inflowing sediment volume for Twengwen Reservoir is 5.6 million m$^3$ per year.
(1) With the newly-constructed desilting tunnel with capacity as 1,000 m$^3$/s, the annual volume by sediment venting during floods is estimated as 1.60 million m$^3$ per year.
(2) The desilting volume of hydro-suction is currently increased to 3.0 million m$^3$ per year.
(3) While several watershed management measures are expected to reduce the inflowing sediment volume by 0.5 million m$^3$ per year, there is still a gap of 0.5 million m$^3$ per year to be desilt by mechanical excavation.
(4) The cost for mechanical excavation is approximately 20 USD per desilt volume (m$^3$) and can be replaced by the empty flushing with manageable incremental water shortage.
(5) More aggressive flushing operations accommodate the uncertainties of effectiveness of watershed management measures, spillage-required sluicing and removal of hydrosuction deposition, and potentially recover the siltation volume of reservoir.
2. The original Figure 18 and Table 5 are updated as Fig.10 and Table 2 in the revised manuscript according to the precise point by the reviewer. |